# Turbulence measurements suggest high rates of new production over the shelf edge in the northeastern North Sea during summer

Jørgen Bendtsen[1], Katherine Richardson[2]

[1]ClimateLab, Symbion Science Park, Copenhagen, 2100 O, Denmark
[2]Center for Macroecology, Evolution and Climate, Natural History Museum of Denmark, University of Copenhagen, Copenhagen, 2100 O, Denmark

*Correspondence to*: Jørgen Bendtsen (jb@climatelab.dk)

**Abstract.** New production, i.e., that driven by allochthonous nutrient inputs, is the only form of primary production that can lead to net increases in organic material and is, therefore, important for understanding energy flow in marine ecosystems. The spatial distribution of new production is generally, however, not well known. Using data collected in July, 2016, we analyse the potential for vertical mixing to support new production in the upper layers of the northeastern portion of the North Sea. Relatively large (up to >0.5 mmol N $m^{-2}$ $d^{-1}$ ) nitrate fluxes due to turbulent vertical mixing into the euphotic zone were found at some stations over the shelf-edge, while low values (< 0.1 mmol N $m^{-2}$ $d^{-1}$) were found in the deeper open area north of the shelf-edge. The low vertical mixing rates (dissipation rates of turbulent kinetic energy below $10^{-8}$ W $kg^{-1}$, corresponding to vertical turbulent diffusion coefficients of $10^{-6}$ - $10^{-5}$ $m^2$ $s^{-1}$) implied f-ratios of < 0.02 in the open waters north of the shelf-edge. In the shallow (< 50 m) southern and central part of the study area, inorganic nutrients were low and nitrate undetectable suggesting negligible new production here, despite relatively high concentrations of chlorophyll a being found in the bottom layer. Thus, high rates of new production seem to be concentrated around the shelf-edge zone and in association with localised features exhibiting enhanced vertical mixing. We find that the nutricline depth is significantly deeper at the shelf edge and interference with increased mixing in this deeper depth range can explain the increased diapycnal nitrate fluxes. Overall, this suggests that the shelf-edge zone may be the major nutrient supplier to the euphotic zone in this area during the period of summer stratification.

## 1 Introduction

New Production (NP), i.e. primary production (PP) based on inorganic nutrients recently introduced to a system, balances net losses of organic material in the euphotic layer when considered over longer periods (Dugdale and Goering, 1967). Net loss of biomass from the surface layer may be due to sinking phytoplankton or transfer to, and subsequent losses from, higher trophic levels such that this cycle between nutrient input, PP and remineralisation of organic matter constitutes an organic biological pump in the upper ocean (Volk and Hoffert, 1987). In addition to NP, total PP is comprised of photosynthesis driven

by nutrients recycled within the system, itself. The ratio of NP to total PP is referred to as the f-ratio (Eppley and Peterson, 1979). This ratio is known to vary both seasonally and spatially.

Given that NP can lead to a net increase of organic material, its occurrence can be expected to have important consequences for food webs and the distribution of marine organisms. Scott et al. (2010) analysed the distribution of marine mammals and seabirds in the northwestern North Sea and they found the distribution of these organisms to be best explained by the presence of a sub-surface chlorophyll maximum (SCM). They furthermore speculated that bathymetric features induced vertical mixing that could bring nutrients to depths containing the chlorophyll peak, which then could lead to localised NP. This potential link between localised increased NP in sub-surface layers and the highest trophic levels may, therefore, be an important feature in ecosystem functioning. The distribution of NP, especially over small spatial scales, is however not well understood. The purpose of this study, therefore, was to describe the potential for NP based on the vertical mixing of nutrients into the euphotic zone to occur in the northeastern North Sea, a highly productive temperate ocean region. The summer season was chosen for the study as this region is thermally stratified at this time and surface waters characterised by low or undetectable concentrations of inorganic nutrients and low phytoplankton biomass. Thus, it is generally assumed that little NP is occurring in these waters at this time.

The northeastern North Sea is characterised by the transition from a relatively shallow southern region (depth ~ 40 m) across a shelf-edge (depth ~80 - 130 m) towards the deep (> 400 m) Norwegian Trench leading into Skagerrak (Fig. 1). The general cyclonic circulation is characterised by inflow from the Atlantic along the shelf-edge, either from mainly wind-driven inflow between the Shetland and Orkney Islands or from transports along the Norwegian Trench (Winther and Johannessen, 2006). Outflow takes place along the Norwegian coast, partly as less saline water masses in the Norwegian Coastal Current, and with an estimated net transport of about 1 Sv in either direction (Danielssen et al., 1990). The relatively large cyclonic flow has also been found in model studies (Winther and Johannessen, 2006; Pätch et al., 2017) where current speeds along the northeastern shelf-edge towards the Skagerrak intensify (Holt and Proctor, 2008) and are found to be associated with a large eddy-activity (Røed and Fossum, 2004). This dynamic area was chosen as the study area as it is a potential location for increased vertical mixing and nutrient transports due to, for example, eddy-activity, breaking internal waves, and upwelling at the shelf-edge.

A large part of the central and northern North Sea becomes stratified during the summer season with an SCM between ~20-40 m depth. Richardson et al. (2000) found that a significant amount of PP occurs at the SCM and argued that much of this could be NP supported by fortnightly tidally driven input of nutrient rich deep water to the SCM. Fernand et al. (2013) found that up to 60% of PP took place in the SCM in the central and northern North Sea during the summer. Those workers also reported a distinct SCM in the northern North Sea where the depth was larger than ~40 m, i.e. areas where tidally induced mixing cannot break the stratified water column. The northern North Sea is relatively productive during the summer season with PP levels of

~0.5 - 0.9 g C m$^{-2}$ d$^{-1}$ (Weston et al, 2005; Fernand et al., 2013). However, few studies have been made of PP at the northern shelf-edge and the distribution of NP is unknown.

Shelf areas, in general, constitute an important link between the large reservoirs of dissolved substances in the open ocean and coastal water masses and shelf processes play an important role in global cycles of nutrients, oxygen and carbon (Gattuso et al., 1998). Transports across the shelf edge bring nutrient-rich deep-water masses into more shallow and well-mixed areas and stimulate production above the shelf. Various cross-shelf transport processes can facilitate this exchange (Brink, 2012), thus different exchange processes may dominate in different shelf-edge regions. However, shelf-edge areas share common dynamical features due to their relatively steep bathymetric slopes and the separation between coastal and open ocean water masses. Therefore, the conditions in the north eastern North Sea may be representative for similar shelf-edge regions not directly exposed to the open ocean basins.

Here, we analyse the physical and biological processes affecting PP and NP during the stratified summer period across the shelf-edge in the northeastern North Sea based on data collected during the VERMIX cruise in July 2016. First, we describe the distribution of water masses, nutrients and vertical mixing in the area. Then, we present results from PP-incubations and, based on averaged representative photosynthetic parameters, calculate PP for the whole study area. Nitrate fluxes and NP are calculated and compared with PP to derive f-ratios. Finally, the implications for NP in the North Sea are discussed.

## 2 Methods

### 2.1 Study area and hydrographic measurements

The VERMIX study covered an area around the shelf-edge in the north eastern North Sea from the shallow shelf area (depth <50 m) across the western part of the Norwegian trench (~500 m) towards the coast of Norway (Fig. 1a). In this study, we define the shelf-edge zone as being where depths are between 80 m and 130 m and refer to areas south and north of this zone as the shallow shelf area (<80 m) and the area off the shelf edge zone (>130m), towards Norway, respectively.

The cruise was carried out on board R/V Dana (Technical University of Denmark) during the period 12 - 31 July 2016. Stations were placed along five transects between 56.2 - 57.8° N and along 8.25° E (Tr1), 7.75° E (Tr2), 7.25° E (Tr3), 6.75° E (Tr4) and 6.25° E (Tr5), respectively (i.e. ~25 km between transects). Wind and light conditions were typical for the period with windy conditions during the first week (along Tr1 and Tr2 with wind speeds up to 20 m s$^{-1}$) and relatively calm conditions during the rest of the cruise (Tr2-5). In total, 132 stations were sampled along the transects with a general distance between stations of 10 km and at sections along Tr2 and Tr4 this distance was decreased down to ~1 km. Measurements along these two transects were made several times during the cruise to examine temporal variability. Measurements over 22 - 36 hours

were made at two time series stations (T1, T2) located at Tr2 and Tr4, respectively. Hydrographic measurements of conductivity, temperature and depth (CTD) were made with a Seabird SBE911 plus system, including two sets of temperature, conductivity and oxygen sensors (SBE43) and equipped with a SCUFA fluorometer, a PAR-sensor (Licor) and 12 Niskin bottles (30 l) for water sampling. All salinity values are reported as practical salinity ($S_P$) and referred to as salinity (S) and TEOS10 (IOC et al., 2010) was applied for calculating properties of seawater.

## 2.2 Measurements of turbulence and micro-structure

Micro-scale turbulence was measured with a loose-tethered free-fall Rockland Scientific International (RSI) VMP-250 microstructure vertical profiler equipped with two shear probes, a FP07 thermistor and a micro-conductivity sensor. Microstructure of temperature and salinity only supported the analysis of shear-induced turbulence and are not discussed further here. The profiler was also equipped with a CT- and pressure-sensor (JFE Advantech). The JFE-CTD data were binned in 0.1 dbar and the conductivity sensor was aligned with temperature by a delay of 0.14 s. Microstructure measurements were made at 512 Hz and stored at 64 Hz and the CT-sensor operated at 16 Hz. Turbulence profiles were obtained while the ship was freely drifting. In total, 253 casts down to about 5 m from the bottom, or to a maximum depth of 200 m above deeper areas, were made during the cruise. Every cast provided two shear-measurements and the dissipation rate ($\varepsilon$) of turbulent kinetic energy (TKE) was calculated with software provided by Rockland Scientific. Dissipation of TKE was estimated from below 10 m depth in order to exclude possible disturbances from the ship (depth of ship was 5.7 m) and analysed in bins of 8 seconds with 50% overlap, corresponding to a resolution of ~3 m, following the method described by Wolk et al. (2002) and revised by Lueck (2016).

The uncertainty and short-term temporal variability of the calculated $\varepsilon$-values were assessed from a time series station (T1) at Tr2 where $\varepsilon$-values were calculated from 107 casts carried out in three sequences over a 22 hours period (Supplementary material). In general, there was good accordance between estimates of $\varepsilon$ obtained from the two sensors and the error-distribution, defined from the difference of the logarithm (to the base 10 and in units of W kg$^{-1}$) between the two estimates, had an absolute standard deviation of 0.14 (Fig. S1). In order for measurements to be included in the analysis, we applied the criterion that the difference between the measurements made by the two sensors on a single profile should be less than three times the absolute deviation. This led only to the removal of a few pairs of $\varepsilon$ from the data set. Short term variation at T1 was also found to be relatively small and temporal changes between subsequent casts were considered to have a little influence on the calculated $\varepsilon$-values (time series analysis in Supplementary material). Therefore, $\varepsilon$-values were, in general, derived from a single cast between the relatively closely spaced stations, and the $\varepsilon$-value obtained by averaging the calculated value from the two shear-probes was reported.

The vertical turbulent diffusion coefficient ($k_v$) was estimated from the dissipation rate of TKE (Osborn, 1980):

$$k_v = \Gamma \, \frac{\varepsilon}{N^2} \tag{1}$$

with a mixing efficiency ($\Gamma$) and the Brunt-Väisälä frequency ($N^2$) determined by linear regression of density ($\rho$) with depth in 10 m intervals:

$$N^2 = -\frac{g}{\rho_0}\frac{\partial \rho}{\partial z} \tag{2}$$

where $\rho_0$ is a reference density (1027 kg m$^{-3}$) and g is the gravitational acceleration.

The mixing efficiency, $\Gamma$, is here assumed constant and equal to 0.2 and this is within the range $\Gamma \leq 0.20$ suggested by Osborn (1980). The value of 0.2 is supported by numerical studies when the turbulent intensity, defined by $Re_b = \varepsilon/(\nu\, N^2)$ where $\nu$ is the molecular viscosity, is within the range 7-100 (Shih et al., 2005), and the model by Bouffard et al. (2013), also validated

against field data, finds the valid range to be between 20 and 400. Outside this mixing regime, the mixing efficiency is less than 0.2. We apply the range of Bouffard et al. (2013) and this implies that our calculated diffusion coefficients with a constant mixing efficiency of 0.2 are valid in the range between $0.5 \cdot 10^{-7}$ - $1.0 \cdot 10^{-4}$ m$^2$ s$^{-1}$ (we use $\nu = 1.2 \cdot 10^{-6}$ m$^2$ s$^{-1}$). This range encompasses the data used for calculating vertical nitrate fluxes in the euphotic zone in our study.

**2.3 Measurements of nutrients and chlorophyll a**

Water samples were taken at CTD stations from standard depths (5, 10, 30, 40, 60, 100, 200 m and close to the bottom). In addition, a sample from the depth (i.e. peak) of the subsurface chlorophyll a maximum (SCM) was sampled. At some stations, a distinct second chlorophyll peak or "extreme" (i.e, a subsurface chlorophyll a extremum, SCE) was observed above the SCM. In some cases, water for photosynthesis incubations and inorganic nutrient (nitrate, nitrite, ammonia, phosphorous and silicate) determination was taken from these SCEs. However, in this study we refer only to subsurface samples from the SCM. Water

samples for nutrient determination were tapped from the Niskin bottles and immediately frozen. The sample was subsequently thawed, filtered (Milipore Millex-GP Hydrophylic PES 0.22 μm), and analysed for nitrite, nitrate, ammonia, phosphorous and silica by wet-chemistry methods according to Grasshoff (Ed.) et al. (1983) (detection limits were 0.04, 0.1, 0.3, 0.06 and 0.2 μM, respectively) at Aarhus University, Department of Bioscience. In total 649 water samples were analysed for nutrients.

Chlorophyll a (GF/F filtered and extracted in 90% ethanol) was determined fluorometrically (Trilogy, Turner Designs) and used to calibrate the rosette mounted fluorometer (chlorophyll a is simply referred to as chlorophyll below). A relatively constant background fluorescence was measured in the deep profiles (>200 m). As we see no reason for at constant distribution of chlorophyll in waters down to depths of ~500 m, we considered this fluorescence to be generated by material other than chlorophyll. Therefore, the fluorescence was averaged between 100-500 m at a deep station on Tr1 (57.83 °N, average value

of 0.023 volt). This relatively small value was treated as an unknown background and subtracted from the fluorescence signal (F). The chlorophyll concentration (chl, mg chl m$^{-3}$) was then determined from a linear regression between fluorescence (F) and filtered chlorophyll samples as chl = 3.23 F - 0.08 ($R^2$ = 0.71, n = 205).

**2.4 Calculation of photosynthetic parameters and PP**

Primary production (PP) was determined from incubations made with Photosynthetic Available Radiation (PAR) at 12 different light intensities from 0 - ~800 µE m$^{-2}$ s$^{-1}$. In total, 64 incubations were made with water sampled from the surface and/or SCM at 41 stations and 23 and 25 incubations were selected to represent the conditions at the surface (5 m) and SCE,

respectively (Table 1). Total PP was then calculated according to Platt et al. (1980):

$$PP(chl(z), PAR(t,z)) = \int_{24h} \int_{-Deu} P_{max}^B(z)\, chl(z) \left(1 - exp\left(-PAR(t,z)\, \frac{\alpha^B(z)}{P_{max}^B(z)}\right)\right) exp\left(-PAR(t,z)\, \frac{\beta^B(z)}{P_{max}^B(z)}\right) \quad dz \quad dt$$

(3)

where the integral is carried out in the vertical (z) from the euphotic depth, Deu, (assumed to be 0.1 % PAR, see below) to the surface and during a 24-hour period (t). The photosynthetic parameters ($P_{max}^B$, $\alpha^B$ and $\beta^B$) were determined by fitting the

photosynthesis (P) vs. E curves generated from the incubations to the PP expression, i.e. the terms in the integral in Eq. (3), by a non-linear Levenberg-Marquardt least-squares routine (Press et al., 1992) and normalised with *in situ* chlorophyll concentration obtained from the calibrated CTD-fluorescence measurement. The case without photo-inhibition, i.e. $\beta = 0$ (Webb et al., 1974), in general, resulted in a poorer representation of the data (except for six incubations where the photo-inhibition term was set to zero) and the incubation experiments were, therefore, analysed with the photo-inhibition term

included in Eq. (3). These model fits were found to be in very good accordance with the incubation data (Fig. S2). The maximum rate of photosynthesis, defined as $P_{max}^{B}{}^{*}$, was calculated for each P vs. E curve (note that $P_{max}^B$ only describes the maximum PP in the case without photo-inhibition).

Irradiation was measured continuously during the cruise and the hourly averaged insolation curve during the day was scaled

to equal the daily averaged photosynthetically available radiation (PAR) measured from the MODIS-Aqua satellite measurements (Frouin et al., 2012) where the daily averaged PAR for July 2016 in the study area (13-28 July, 6.2° - 8.3° E, 56.2° - 58° N) was 506 µE m$^{-2}$ s$^{-1}$. The vertical light extinction coefficient ($k_d$) was determined by linear regression of the log-transformed PAR-measurements from each CTD-cast. The average value from all the CTD-casts was found to be $k_d = 0.14 \pm 0.03$ m$^{-1}$ (std.dev, n=177), corresponding to a 0.1 % PAR depth level of 50 m. The 0.1 % PAR depth level was in this study

assumed to represent the bottom of the euphotic zone.

Primary production estimates at individual sites are dependent upon the value for maximum rate of photosynthesis ($P_{max}^B$). However, $P_{max}^B$ (and all other measured photosynthetic parameters) represents the physiological condition of the phytoplankton community at the time of sampling. This means that $P_{max}^B$ may vary as a function of time of sampling

(Richardson et al., 2017) or during different light conditions (e.g., photo-inhibition). Normalisation of the photosynthetic parameters with chlorophyll also represents an uncertainty in the PP estimates at individual stations as, for example, division with low chlorophyll values (e.g. some surface values were ~0.1 mg chl m$^{-3}$) may result in large uncertainty of the normalised

values due to relatively large absolute errors. This uncertainty error has been shown to potentially have a significant impact on the estimation of photosynthetic parameters (e.g., Kumari, 2005; McKee et al., 2015). Finally, the fact that photosynthetic parameters were determined from incubations carried out on only one water sample from each sampling depth represents a source of uncertainty with respect to the estimates of PP at individual stations. Therefore, in order to compare PP estimates from the stations we sampled, we applied average values (median for all stations) of photosynthetic parameters in the surface layer (5 m) and in the SCM in the calculation of PP. The uncertainties associated with the photosynthetic parameters are further considered in the Discussion. Surface values were assumed to represent the photosynthetic parameters in the upper 10 m and average values from the SCM were assumed to represent the parameters for the water column below 10 m.

## 2.5 Vertical nutrient fluxes and f-ratios

Nitrate was depleted in the surface layer at almost every station and, therefore, assumed to be the limiting nutrient. This assumption was supported by the distribution of other nutrients (shown below). The nutricline was defined as the depth where nitrate equalled 0.5 µmol kg$^{-1}$ (the chosen threshold value had, in general, only a minor influence on the nutricline depth because nutrients were so depleted in the surface layer). In general, water samples were taken close to the nutricline, i.e. within +/- 10 m. However, at 5 stations where the nutricline was not resolved well by the sampling depths, samples from the neighbouring station (i.e. within 5-10 km) were included to increase the resolution. To minimise the uncertainty of the nutricline depth estimate associated with the linear interpolation between two neighbouring water samples, the nutricline depth was found by linear interpolation between the corresponding potential density anomalies ($\sigma_\theta$) of the sample depths, and the nutricline depth was then identified from the corresponding $\sigma_\theta$ in the CTD-profile. This approach is based on the assumption that the nitrate concentrations between two water samples are more closely related to water mass characteristics than linearly to depth changes, i.e. a sharp pycnocline, not resolved by the water samples, is taken into account when the nutricline depth is estimated by this method. A comparison was made for the whole data set between nutricline depth estimates based on linear interpolation using depth and potential density anomaly levels, respectively. The difference between the two methods averaged -1.4 ± 2.5 m (std. dev., n=77) and the differences ranged between -9.9 - 2.4 m. Thus, in general, the depth level-based method resulted in a deeper nutricline (-1.4 m) than the applied $\sigma_\theta$-method.

At all stations where nitrate was present (n=73), the nutricline was found to be located above the depth of the 0.1% PAR light level and, therefore, the flux of nitrate ($F_{NO3}$) for NP was found by calculating the maximum flux into the euphotic zone (only transports from vertical turbulent mixing were measured, thus, contributions from vertical advection were not considered). The vertical nitrate flux ($F_{NO3}$) due to turbulent mixing is defined as:

$$F_{NO3} = -k_v \frac{\partial NO_3}{\partial z} = -k_v \frac{\partial NO_3}{\partial \rho} \frac{\partial \rho}{\partial z} \tag{4}$$

where the vertical nitrate gradient is reformulated in terms of density (Sharples et al., 2007). By applying this reformulated relationship, together with the definition of the vertical turbulent diffusion coefficient in Eq. (1), an expression for the nitrate flux in terms of the measured dissipation rate of TKE becomes:

$$F_{NO3} = \frac{\Gamma \, \varepsilon \, \rho_0}{g} \frac{\partial NO_3}{\partial \rho} \qquad (5)$$

The advantage of this expression, compared to simply applying $k_v$ and the vertical nitrate gradient directly, is the dependence on the measured dissipation rate, rather than the derived $k_v$-value. The latter includes the calculation of the Brunt-Väisälä frequency, cf. Eq. (2). This term becomes close to zero in very turbulent conditions and, thereby, $k_v$ becomes sensitive to rounding errors and other uncertainties in the measurements. The gradient of nitrate at a given depth was determined linearly from the nearest water sample depths.

PP was related to the vertical nitrate flux by estimating the f-ratio, calculated as: $f = \eta_{N:C} \, F_{NO3}/PP$, where $\eta_{N:C}$ is the Redfield ratio between nitrogen and carbon, i.e. 16:106 (Redfield et al., 1963).

## 3 Results

### 3.1 Water mass distributions

The northern North Sea is a region in which North Atlantic water transported towards the Skagerrak, water masses formed in the shallower part of the North Sea and outflowing low-saline water from the Baltic Sea come together. Water masses and the general circulation in the area have been described by Danielssen et al. (1990), and the general water mass characteristics they identified are included in the description below. Their analysis was based on measurements obtained in the Skagerrak and the northeastern North Sea in 1990-91.

The inflow of Atlantic water to the northern North Sea, in general, takes place between the Orkneys and Shetland, above the shelf east of the Shetland and in the deep Norwegian channel and the deep inflow in the Norwegian channel dominates during the summer period (Winther and Johannesen, 2006). Atlantic water masses (AW) are characterised by salinities > 35 and can be further delineated as upper Atlantic water (AWu, S = 35.00 - 35.15 and T = 8 - 10 °C) and a slightly colder and more saline

deeper Atlantic water mass (AWd, S = 35.15 - 35.32, T = ~7 - 10 °C, Fig. 1b). The AWd was observed on the westernmost transects (Tr4 and Tr5) between 100 - 300 m along the shelf edge. High saline AWd was also present in the depth range 120 - 200 m on Tr1 (not shown). AWu was observed at the shelf edge at depths below 60 m on Tr2 - 5 whereas it was found at depths below 80 m on Tr1 (Fig. 2). An upward doming of the AWu was seen to reach to ~ 30 m between the shelf-edge and the Norwegian coast from where isotherms and isohalines tended to deepen towards Norway. Central North Sea water (CNSW,

S = 34.80 - 35.0 and T = 8 - 10 °C), a mixture of Atlantic water and Scottish coastal water, was located above AWu and a

distinct frontal zone between the two water masses was seen along the shelf-edge where bottom depths ranged between 40 and 60 m.

The influence of southern North Sea water (SNSW, S = 34.50 - 34.8, T = 8 - 12 °C) (note the larger temperature interval in mid-summer than defined in Danielssen et al. (1990)) and low nitrate concentrations, originating from the English Channel, was seen at the shallower stations (< 40 m) on all transects except Tr1. Warm subsurface water between 20 - 40 m at Tr1 indicated an influence from the Jutland coastal water mass (JCW, S = 32 - 34 and T = 10 - 15 °C) on this transect. The surface mixed layer varied between 5 - 15 m with the lowest salinities seen in the Norwegian coastal water mass (S < 28) and the highest surface salinities being found above the well mixed shallow parts of the North Sea (S ~ 34.5).

A distinct subsurface oxygen maximum ($O_2$ > 240 µmol $kg^{-1}$) between ~15 - 35 m depth characterised the deeper area north of the shelf-edge whereas a less well-defined maximum was present above the shelf-edge and in the shallow North Sea (Fig. 2c). The bottom water (~20 - 60 m) in the shallow areas exhibited relatively low oxygen concentrations, i.e., < 200 µmol $kg^{-1}$. High nitrate concentrations characterised Atlantic water masses whereas very low nitrate concentrations were observed in the surface layer and above the shallow southern part of the area (Fig. 2d).

## 3.2 Chlorophyll and nutrient distributions

The subsurface chlorophyll concentrations were relatively high (> 2 mg chl a $m^{-3}$) in the shallower regions of the study area (Fig. 3). A narrow well-defined subsurface chlorophyll maximum extended from the shelf-edge towards the Norwegian coast, i.e. close to the 27 kg $m^{-3}$ isopycnal. The depth of the subsurface chlorophyll maximum tracked the nutricline from the shelf-edge, where the nutricline was separating the nutrient-depleted surface layer from the nutrient rich Atlantic water (Fig. 4a, b).

In general, a sharp nutricline separated the nutrient depleted surface layer from the nutrient rich Atlantic water with nitrate concentrations above 6 µmol $kg^{-1}$. For example, an increase from values below the detection limit (0.1 µmol nitrate $kg^{-1}$) to values above 2 µmol $kg^{-1}$ was observed across a 5 m distance (35 - 40 m depth) at the shelf-edge on time series station T2 at Tr4 (57.31 °N). The nutricline between CNSW and AWu was located at density-anomalies of ~27.3 kg $m^{-3}$, whereas the nutricline was located below the 26.5 kg $m^{-3}$ isopycnal in areas influenced by SSW and Norwegian coastal water masses. Thus, high nutrient concentrations were found in the cold (< 8 °C) Atlantic water masses and low concentrations were associated with CNSW, SNSW and SSW.

Nitrate was not detectable at the southernmost stations south of 56.88° N at Tr5, 57.00° N at Tr4 and Tr3 and 57.21° N at Tr2, whereas the southernmost station at Tr1 showed detectable nitrate (0.2 - 0.3 µmol $kg^{-1}$ between the surface and 30 m) at the near-coastal station, likely influenced by nutrient-rich JCW (Fig. 1). Bottom samples were typically made down to ~5 m from

the bottom, i.e. within the benthic boundary layer, and, considering the relatively large turbulent diffusion coefficients in the benthic boundary layer, this supported the interpretation that no detectable nitrate was present in the water column at the southernmost stations along Tr2-Tr5.

The assumption that nitrate was the limiting nutrient for phytoplankton was supported by the phosphate distributions, in that the N:P-ratio was found to be significantly below Redfield (16:1) at all stations and all depths, with the exception of a few samples above the shallow area (Fig. 4c). Nitrate depletion was observed at stations above the shallow southern North Sea, where excess phosphate compared to the Redfield N:P ratio suggested significant denitrification to be occurring. There was, in general, no indication of silicate limitation because relatively high values (up to 10 µmol kg$^{-1}$) were observed above the
nitrate-depleted shallow area and low silicate values in the deeper part were also associated with low nitrate concentrations (not shown).

### 3.3 Mixing and vertical nitrate fluxes

Dissipation of TKE was low, i.e., $< 10^{-9}$ W kg$^{-1}$, below ~40 m in the deeper areas north of the shelf-edge but increased by an
order of magnitude above the shelf-edge, where values up to ~$10^{-6}$ W kg$^{-1}$ in the benthic boundary layer were recorded (Fig. 5a, b). Dissipation rates of TKE in the 20 – 40 m depth range varied between $10^{-8}$ - $10^{-9}$ W kg$^{-1}$ above the deeper areas and increased to $10^{-7}$ - $10^{-8}$ W kg$^{-1}$ above the shallow areas along Tr2. In the upper part of the water column, between 10 - 20 m depth, the dissipation rate of TKE increased to $10^{-5}$ - $10^{-7}$ W kg$^{-1}$ due to mixing induced by wind and waves. The distribution of the vertical turbulent diffusion coefficient showed a characteristic pattern along all transects where very low $k_v$-values of
$<10^{-6}$ m$^2$ s$^{-1}$ between 15 - 30 m depth characterised the deeper open areas, i.e. around the pycnocline. Increased mixing was seen at the shelf-edge and in the ~10 m thick benthic boundary layer further south above the shallow shelf (Fig. 5c,d; note that the shallow turbulence profiles on Tr4 did not resolve the benthic boundary layer). The highest $k_v$-values, i.e., $> 10^{-4}$ m$^2$ s$^{-1}$, were observed at the shelf edge and in the benthic boundary layer above the shallow part of the shelf.

Vertical nitrate fluxes were calculated for all stations where both nutrients and turbulence profiles were measured. Examples are shown for four stations across the shelf edge on Tr4, obtained 27 July between 14h and 17:30h, with a station spacing of ~5km, in an area where there is a northward increase of the depth from 82 to 122 m (Fig. 6). The nutricline was located at ~35 m and this depth was relatively close to the increased mixing in the bottom boundary layer. Increased mixing is seen at the station located at 57.404 °N (Fig. 6c, g) and this results here in a maximum nitrate flux to the euphotic layer of 0.3 mmol N m$^{-}$
$^2$ d$^{-1}$ at ~40 m. In general, the maximum vertical nitrate flux into the euphotic zone in the deeper parts along the five transects was located 5-10 m below the nutricline (Fig. 3, orange squares). whereas the SCM was located at the nutricline (Figs. 3, 6a). Although the nitrate flux at this depth was relatively low, this showed the close relationship between SCM and nutricline depth.

### 3.4 Photosynthetic parameters

Photosynthetic parameters can vary as a function of sampling time and in situ light conditions. Therefore, in order to be able to compare the potential PP at different stations, we averaged photosynthetic parameters for the surface layer (5 m) and from the SCM, respectively. In total, 64 incubations were made. Fifty eight of the incubations were fitted with the photoinhibition term whereas 6 incubations did not show any significant decrease for increasing light levels. Therefore, the inhibition term was excluded in these calculations. To make the data set representative for the whole study area, incubations from the closely spaced (1-3 km) stations on Tr2 were excluded (i.e. 14 stations between the two stations shown with incubations at Tr2 in Fig. 1a) and, to reduce the impact from outliers, representative values are calculated as median and median absolute deviation values (Table 1).

The median chlorophyll concentrations at the surface and SCM were 0.16 and 1.68 mg chl a m$^{-3}$, respectively. This reflected the general increase in chlorophyll with depth observed over the entire area. The chlorophyll-normalised photosynthetic parameters of the photosynthetic rate constant ($P^B_{max}$) and the slope of the PE-curve ($\alpha^B$) in the surface layer were 5.48 µg C (µg Chl h)$^{-1}$ and 0.041 µg C (µg Chl h µE m$^{-2}$ s$^{-1}$)$^{-1}$, respectively, and the corresponding values of $P^B_{max}$ and $\alpha^B$ at the SCM were 2.33 and 0.027. Thus, $P^B_{max}$ was significantly lower at the SCM, in general accordance with previous studies (see review in Richardson et al., 2016) whereas $\alpha^B$ showed a weak decrease with depth (with overlapping uncertainty intervals between the two depth levels). In general, $\alpha^B$ has been found to increase significantly with depth (resulting in a more efficient photosynthetic response at low light levels) and inspection of the vertical distribution showed at tendency to higher values between 15 -25 m depth (i.e. 2 - 3 e-folding depth of PAR) and lower values below 30 m resulting in a lower median value from SCM level. It was noted that a similar pattern of $\alpha^B$, with a subsurface maximum, has been observed at the European shelf in the Celtic Sea (Hickman et al., 2012).

There was a significant averaged photoinhibition ($\beta^B$) at both depth levels (1.70 and 3.00 (10$^{-3}$ µg C (µE m$^{-2}$ s$^{-1}$ h)$^{-1}$)). However, the average values at these depths were not significantly different. The maximum PP was characterised by $P^B_{max}{}^{*}$-values of 4.76 and 1.72 µg C (µg Chl h)$^{-1}$ at the surface and SCM, respectively, corresponding to maximum PP at PAR-levels of 413 and 192 µE m$^{-2}$ s$^{-1}$. The averaged parameters for the two depth levels of $P^B_{max}$, $\alpha^B$ and $\beta^B$ were applied for calculating PP in the water column according to Eq. (3). A comparison was made between the resulting PE-curves and the maximum averaged PP from all the incubations (i.e. the $P^B_{max}{}^{*}$ and $E_{max}$ in Table 1). These values were found to fit the averaged curves for the two depth levels within 5% and, therefore, the PE-curve based values in Table 1 were assumed to be representative of average conditions and applied in the calculation of PP for all of the stations.

## 3.5 Vertically integrated chlorophyll, PP and NP

The vertically integrated chlorophyll in the euphotic zone (50 m) showed a local maximum south of the shelf-edge along all transects and a decrease in chlorophyll further south at the two westernmost transects, i.e. Tr4 and Tr5 (Fig. 7a). Relatively low values were observed in the area north of the shelf-edge with the exception of an area close to Norway where high values were observed. The averaged vertically integrated chlorophyll from all stations was $29.9 \pm 7.8$ mg chl a m$^{-2}$ (std. dev., n = 128).

The distribution of vertically integrated total PP showed a similar pattern with increased PP-levels south of the shelf edge and a somewhat reduced PP further south at the two westernmost transects (Fig. 7b). A tendency to minimum values was observed at the shelf edge with an increasing tendency towards Norway with high PP-values being observed at a few stations near the Norwegian coast. The averaged vertically integrated PP from all stations was $476 \pm 110$ mg C m$^{-2}$ d$^{-1}$ (n=128).

The maximum nitrate flux into the euphotic zone ($F_{NO3}$) and the f-ratio were calculated for each station where nitrate and turbulence measurements were made and where nitrate was present in detectable concentrations. The vertical distribution of the maximum nitrate flux was, in general, characterised by being located below the SCM (cf. Fig. 3, Table 2). However, the horizontal distribution showed a characteristic pattern where increased fluxes and f-ratios were found along the shelf-edge on Tr2-Tr5 (nitrate fluxes were not measured at the shelf edge and at shallow stations at Tr1). Increased fluxes were also observed near the Norwegian coast at Tr1 and Tr4, and low nitrate fluxes and f-ratios characterised the open area between the shelf edge and Norway (Fig. 7c, d).

## 3.6 Distributions across the shelf-edge

A comparison of distributions of PP related parameters across the shelf-edge showed common trends over all five transects (Fig. 7, S3). Vertically integrated chlorophyll and PP were relatively low, with values of ~20 mg chl a m$^{-2}$ and ~400 mg C m$^{-2}$ d$^{-1}$, respectively, on the southernmost parts of the transects. A gradual northward increase towards maximum values (of ~60 mg chl m$^{-2}$ and >1000 mg C m$^{-2}$ d$^{-1}$) was observed within a distance of ~20 km south of the shelf edge on Tr3 - Tr5. Transect 2 did not reach as far south of the shelf-edge. This could explain the southern maximum in chlorophyll and PP at Tr2. Transect 1 was influenced by nutrient rich JCW and this may be the reason for the high chlorophyll concentration (43 mg chl a m$^{-2}$) and PP (1746 mg C m$^{-2}$ d$^{-1}$) at the southernmost station on this transect.

Relatively low values of ~20 mg chl m$^{-2}$ and 400 mg C m$^{-2}$ d$^{-1}$, respectively, characterised chlorophyll and PP at the shelf-edge zone. These values increased slightly towards Norway to ~30 mg chl a m$^{-2}$ and 5-600 mg C m$^{-2}$ d$^{-1}$, respectively. High values were observed in Norwegian coastal water masses on Tr1 and Tr4, probably due to PP being stimulated by coastal upwelling (also indicated by relatively cold Norwegian coastal surface water masses observed from satellite in Fig. 1b). Stations on Tr2

and Tr4 were visited twice within one week. Significant temporal variation of chlorophyll and PP was observed (Fig S3) between samplings. Chlorophyll increased by up to 40% above the shallow area near the shelf edge and a similar increase was seen in PP. Relatively large temporal variations were also seen in these parameters in the coastal Norwegian water masses.

A nutricline could, in general, not be identified above the shallow areas south of the shelf-edge except at two stations on Tr3 and Tr4 (cf. Fig. 2c, d, at both stations high nitrate concentrations of 0.9 - 1 µmol kg$^{-1}$ were only measured at 5 m while low values of 0.1 - 0.3 µmol kg$^{-1}$ were found below 10 m). The deepest nutricline depths (~40 m) were found near the shelf-edge. Nutricline depth decreased to ~20 m going northwards but then increased again in the coastal Norwegian water masses (Fig 8c). Nitrate fluxes were generally very low ($< 0.1$ mmol N m$^{-2}$ d$^{-1}$) in the deeper area north of the shelf edge due to low vertical

mixing in the upper 50 m. However, the largest nutrient fluxes were seen in the shelf edge zone and near the Norwegian coast. This resulted in f-ratios above 0.10 in these regions compared to $< 0.02$ for the remainder of the study area (Fig. 7, S3).

The distributions were analysed across the shelf edge by dividing the stations into three depth ranges characterising the shallow area (50 - 80 m), the shelf edge zone (80 -130 m) and the deep area ($> 130$ m), respectively (Table 2). Although the shelf edge

was characterised by the largest nutrient fluxes, the averaged values were not significantly higher than observed above the deeper areas. However, the depth of the maximum flux was found to be significantly deeper (~43 m) above the shelf edge than in the deeper area (~32 m). This can be explained by the significantly deeper nutricline at the shelf edge (~35 m) than observed above the deeper area (~27 m). Distributions of vertically integrated chlorophyll and PP support that minimum values are found above the shelf edge. However, the low values are not significantly different from the larger values above the deeper

and shallower part of the area.

### 3.7 Temporal variability at the shelf edge

Data was collected over 36 hours from a time series station located on the shelf edge on Tr4 (depth 82 m, 57.314 °N, 6.765 °E, from 29 July 7:07h to 30 July 19:25h) where CTD-casts and turbulence measurements were carried out every hour

(although no turbulence measurements were made during the night at this station). Variability was related to the tidal current obtained from the barotropic OTIS model of the North Sea (Egbert and Erofeeva, 2002). The model depth (h) at the time series location was 80 m, it considers eleven tidal constituents, its general performance has previously been validated and a comparison was also made against tide tables for the period from Hanstholm harbor located on the Danish coast (57.17 °N, 8.62 °E). The model result showed excellent agreement with the tidal phases. Tidal energy input was estimated from the cube

of the barotropic tidal current ($u^3$, e.g. Simpson and Sharples, 2012). This energy input represents the source for tidally induced vertical mixing and can be compared with turbulent mixing in the water column.

The water column was characterised by a homogenous mixed layer in the upper 5-10 m, a pycnocline at ~20 m, and a relatively homogenous temperature and salinity distribution below ~45 m ($\theta < 8$ °C) (Fig. 8). A sharp nutricline between 38 - 40 m

separated the nitrate-depleted surface layer from the nitrate rich bottom water (~ 2 µmol kg$^{-1}$, not shown). An oxygen maximum layer between 20 - 40 m separated the oxygenated surface layer from the bottom layer with a relatively lower oxygen concentration. Weak temperature and salinity stratification was observed in the start and end of the period (i.e. between DoY 210.3-210.7 and 211.3-211.7) and a corresponding change was observed in the bottom oxygen concentration. The period

between DoY 210.6-211.2 was characterised by a relatively homogenous bottom layer (< 40 m). Increased turbulence ($\varepsilon$, $k_v$) and decreased stratification ($N^2$) were also observed in this period (Fig. 8d-h). Mixing in the bottom layer increased simultaneously with tidal energy input (Fig. 8a, b). This indicated that tidal barotropic currents overlain on baroclinic currents along the shelf edge could explain the temporal variability in turbulence.

Dissipation of TKE in the bottom layer at the time of the maximum observed dissipation (DoY=210.8) was comparable with the energy input from the tides (~$10^{-3}$ W m$^{-2}$). Additional energy for turbulence may also be provided from non-tidal currents along the shelf-edge (strong eastward currents below 20 m were noted at the time of some of the turbulence profiles). A relatively large deepening of both T, S and O$_2$ below the pycnocline was observed between DoY ~ 211.25-211.4. Nutrient concentrations below 60 m showed an increase from 2 - 3 to 5 - 6 µmol kg$^{-1}$. This indicated advection of water below the

pycnocline could explain the short-term variation observed.

Thus, the observed temporal variability in water column structure and nutrient distributions at the shelf edge shows that varying mixing intensities may interfere with the bottom of the euphotic zone and, thereby, promote diapycnal nutrient fluxes and stimulate NP. The variability arises due to different physical processes, which cannot be identified in detail from the present

data set. This may also explain the observed variability between stations located in the shelf edge area (Table 2). Periods of low mixing in the euphotic zone may be followed by short periods of intense mixing. Such temporal variability is a challenge to document over a larger area.

## 4 Discussion

The shelf edge was identified in this study as a potential area of localised NP during the stratified summer period in the

northeastern North Sea. Very low nitrate fluxes and f-ratios were estimated for most of the open water extending northwards from the shelf edge towards Norway. Likewise, little or no NP was estimated to be occurring in the nitrate depleted shallow reaches of the southern North Sea. Previous studies have shown that shelf edge areas can be productive regions, for example characterised by increased fishing activity (Sharples et al., 2013). In relation to the observations reported by Sharples et al., we speculate that the abundance of fish in this area could be related to a localised increase in NP relative to surrounding waters.

It can also be noted that the northeastern shelf edge area is characterised by increased fish species diversity (ICES, 2008), suggesting that the introduction of new nutrients to the euphotic region of the water column identified for this region may be influencing both food webs and ecosystem structure.

.

Localised NP at the shelf edge will also influence the oxygen concentration in subsurface water masses (Fig. 2). This can have a direct influence on ambient water with relatively low oxygen above the shallow shelf and influence conditions further downstream, where water masses from the shelf edge eventually reach the North Sea/Baltic Sea transition zone and contribute to bottom water ventilation in more eutrophic areas (Bendtsen et al., 2009).

The possible occurrence of localised NP also implies that physical processes on relatively small spatial scales (~1 km) are important for modelling NP in the area. This is in general accordance with model studies where the resolution of physical processes at high spatial scales was found to be necessary to explain observed patterns of PP in the North Sea (Skogen and Moll, 2000). Holt et al. (2012) applied a high-resolution model of the whole North Sea area and showed that nutrient transports towards the shelf area were of primary importance for understanding PP and also for assessing the impact from climate change. Localised NP at the shelf edge in the northeastern North Sea may be representative for the open ocean - shallow shelf exchange of nutrients more generally. If so, then this NP may also impact nutrient conditions above the shallow central North Sea and may explain some of the recent decadal decline of PP in this area (Capuzzo et al., 2018).

## 4.1 Regenerated production above the shallow shelf

There was no measurable nitrate in the water column at stations south of the shelf-edge., i.e. an area where the whole water column was within the euphotic zone, i.e. < 50 m (Fig. 1a, yellow lines). This indicates that nitrate sinks, i.e. biological consumption or denitrification, exceed nitrate sources, i.e. nitrification, nitrogen-fixation or advective supply during summer months in this region. Significant nitrification may take place in the water column (Clark et al. 2008; Zehr and Kudela, 2011) or in the sediment and, as pointed out in the work of Dugdale and Goering (1967), nitrate can, in this case, not be considered as a non-regenerated nutrient form in the euphotic zone. Yool et al. (2007) estimated a global specific nitrification rate of 0.2 $d^{-1}$, thus, even small concentrations of ammonium could lead to significant nitrification rates. Concentrations of ammonium were, however, very low in the nitrate depleted area. For example, ammonium was undetectable at all nitrate-depleted stations on Tr5 and this indicated a relatively small contribution from nitrification in the euphotic zone. Observed nitrification rates span a large range (Yool et al., 2007) and the importance of nitrification varies between ocean regions (Clark et al, 2008; Fawcett et al., 2015). Therefore, it remains an open question whether or not significant nitrification takes place in the euphotic zone in the area.

Diazotrophy constitutes another potential source of nitrogen to the nitrate-depleted surface layer and $N_2$-fixation has been measured in the southern North Sea at Dogger Bank (Fan et al., 2015). However, the estimated nitrogen fluxes from $N_2$-fixation were very small compared to the relatively high PP. Increased nutrient fluxes and NP at the shelf-edge could also support PP

in the ambient shallow areas through isopycnal transport of dissolved organic material. A substantial fraction of dissolved organic material remineralises on time-scales of days to weeks (Bendtsen et al., 2015; Hansen and Bendtsen, 2014) and isopycnal transport of organic matter could then supply organic nitrogen from the shelf edge zone. Both the nitrification and isopycnal supply of organic material, thus, potentially provide new nitrogen to the shallow area and this confuses the concept of nitrate-based new versus regenerated production here. However, nitrification in the euphotic zone would be based on regenerated production and, therefore, we consider PP to be regenerated production in areas without detectable nitrate in the water column.

Recycling of organic matter above the shallow shelf could, thus, be maintained by regenerated organic matter in the water column or sediment and we analysed whether this was in accordance with estimated carbon and nitrogen pools. The high biomass in the shallow area, indicated by chlorophyll concentrations of ~2 - 4 mg chl m$^{-3}$ in the bottom layer, would rapidly consume a regenerated pool of inorganic nitrogen. The regenerated cycling could, in principle, originate from the winter concentration of nitrate of ~6 mmol DIN m$^{-3}$ in the area (Pätch and Kühn, 2008). The nitrate distribution in the shallow area indicates a large denitrification ranging within 0 - 4 mmol DIN m$^{-3}$ (Fig. 4c), and this would correspond to a pool of more than ~2 mmol DIN m$^{-3}$ available for establishing the phytoplankton biomass. Denitrification rates in the sediment of 0.02 - 0.1 mmol N m$^{-2}$ d$^{-1}$ have been observed at the Dogger Bank between May and August (Fan et al., 2015) and such high rates could explain the indicated nitrate sink. Assuming a Redfield C:N molar ratio of 6.6 and a C:chlorophyll ratio of 1 mg C/50 mg chl, this would equal ~3 mg chl m$^{-3}$, i.e., in general accordance with the observed concentrations of <1 and ~4 mg chl m$^{-3}$ in the surface and bottom layers, respectively. Relatively low oxygen concentrations in the bottom layer also suggested an active bacterial respiration (Fig. 2c) to be occurring here. Thus, regenerated production could be maintained and explained by recycling of an initial nutrient pool from early spring.

### 4.2 Photosynthetic parameters and PP

Estimates of PP based on photosynthetic parameters from $^{14}$C-incubations and in situ conditions of light and chlorophyll rely on several critical assumptions, including the values used for photosynthetic parameters, the distribution of chlorophyll, light conditions and nutrient-carbon relationships. Photosynthetic parameters from individual incubation experiments were well described by the PP-model (i.e. PP described by the terms in the integrals in Eq. (3), where parameter values are taken from incubations, see Fig. S2 for examples of incubation data) so uncertainty is mainly related to the spatial and temporal variability of these parameters. Photosynthetic parameters are bulk parameters describing the physiological response of the phytoplankton community as a whole to a given a photon flux. Thus, both algae composition and actual fitness of the cells contribute to the observed range of these parameters.

The photosynthetic parameters represent the conditions of the phytoplankton community at the time of sampling. However, these may vary during the day (Richardson et al., 2017) or during different light conditions (e.g., photo-inhibition). Thus, PP-calculations based on spatial and temporal averaging of the photosynthetic parameters from the surface and SCM (cf. Table 1) may result in a more representative PP than obtained from photosynthetic parameters obtained from a single PP-incubation.

The values reported here (Table 1) for photosynthetic parameters are comparable with those found in previous studies in the North Sea (e.g. Weston et al., 2005). Variability of the photosynthetic parameters, e.g. the uncertainty of the surface value of $P^B_{max}{}^*$ of ~$\pm 30\%$, implies a corresponding uncertainty in PP. Thus, using a common set of photosynthetic parameters implies an uncertainty even within this relatively small study area, and similar limitations would probably apply to other PP estimates

10 for the North Sea area. These considerations illustrate that a better understanding of the distributions of photosynthetic parameters and the factors underlying these distributions is a prerequisite for improved estimates of PP on regional and larger scales. Given the uncertainties related to the determination of the absolute rate of PP at individual stations, we believe the most robust manner by which to compare PP and NP over our study area is to use constant (average) values for photosynthetic parameters for all stations.

### 4.3 Nutrient fluxes and NP at the shelf edge zone

The largest vertical nitrate fluxes of up to >0.5 mmol N m$^{-2}$ d$^{-1}$
were restricted to the area above the shelf edge where increased mixing in the bottom layer intersected the nutricline within the euphotic zone. Large nitrate fluxes have been observed along the European shelf where tidally induced (Sharples et al.,

20 2001; located in the English Channel) or wind-induced mixing (Williams et al., 2013; Celtic Sea) leads to daily averaged vertical fluxes of 1 - 2 mmol N m$^{-2}$ d$^{-1}$. Significantly larger fluxes have also been observed above steep bathymetric gradients (Tweddle et al., 2013). Wind and tidal mixing may also provide energy for intensified mixing along the northeastern shelf edge in the North Sea, although the present study did not resolve the specific cause for increased mixing. Shelf-edge zones are dynamic areas where cross-shelf exchange may occur due to many different dynamic processes (Huthnance, 1995; Brink,

25 2012).

Mixing associated with wind and tides (e.g. Burchard and Rippeth, 2009) as well as breaking internal waves (Sharples et al, 2007; 2009) has been shown to be important for vertical nutrient fluxes in shelf areas. The specific physical processes behind increased turbulent mixing cannot be identified from the present data set. Measurements on the time series station at the shelf

30 edge showed that increased mixing occurred in phase with the tidal energy input but also that additional energy sources likely contributed to the increased mixing, e.g. energy from non-tidal currents. Short term variability associated with advection of ambient water masses was also observed. This could possibly be related to sub-mesoscale eddies or other transport processes

occurring below the pycnocline. The time series station T2 at Tr4 showed an important feature where mixing associated with the bottom boundary layer increased and intersected the bottom of the euphotic zone. Thus, the combined effect from a deep nutricline and increased mixing provide a mechanism for increased diapycnal nutrient fluxes along the shelf edge.

Other processes may be important on the northeastern shelf edge due to the large in- and outflow of Atlantic water masses. It is interesting to note that the deepening of the nutricline from the open areas towards the shelf edge mirrors the slope of the pycnocline. The slope of the pycnocline implies, via the thermal wind equations, an increased eastward baroclinic velocity component with depth, in accordance with the general cyclonic circulation in the area. Thus, deepening of the nutricline can potentially be explained as a dynamic response to a shelf-edge current transporting Atlantic water into the area.

The conditions across the shelf edge observed during the stratified summer season may, therefore, be considered as a stable quasi-stationary system where mixing at the shelf edge, associated with a deep nutricline and nutrient rich bottom currents, provide nutrients to the euphotic zone stimulating localised new production in this area (Fig. 9). Isopycnal mixing may provide organic matter for increased regenerated production above the nitrate-depleted shallow shelf area and also, together with

upwelling along the Norwegian coast, provide fresh organic material to the euphotic zone off the shelf edge towards Norway. Thus, the tendency towards increased chlorophyll concentrations and PP on either side of the shelf edge might be explained as being a result of the gradual build-up of biomass as nutrients are transported away from the shelf-edge region by isopycnal mixing. Alternatively, the tendency to low values above the shelf edge could also be explained by a larger grazing pressure above the shelf edge. Thus, a full explanation of the tendency to low chlorophyll and PP above the shelf edge area cannot be

determined from these data.

### 4.4 Vertical nutrient fluxes in the euphotic zone

The position of the SCM was closely related to the depth of the nutricline (e.g. Fig. 3 and 6) and located in the middle of the euphotic zone (average nutricline depth from all stations was 29.1 ± 9.7 m, n = 83). The maximum nitrate flux was found at

depth levels between 3.4 m above and 23.1 m below the nutricline. However, on average, it was located 6.4 ± 7.7 m (n=73) below the nutricline depth . Thus, the maximum nitrate flux was within the euphotic zone but, in general, significantly below the SCM (Fig. 3). This implies that internal recycling from below the SCM towards the surface of regenerated nitrogen is necessary for maintaining the phytoplankton biomass in the upper layer. Considering the relatively low vertical mixing rates around the SCM (e.g., a turbulent diffusion coefficient of $<5 \cdot 10^{-5}$ m$^2$ s$^{-1}$ implies a time scale of ~3 weeks for mixing across a

10 m thick layer, Fig. 5), this indicates that other transport processes within the euphotic zone are important in this area. One possible mechanism could be diel vertical migration of plankton. Raven and Richardson (1984) showed the potential for diel vertical migration as an efficient strategy for phytoplankton to get access to both nutrients and light. Such a strategy might be

particularly beneficial for phytoplankton in the low-mixing zone in the pycnocline north of the shelf edge (e.g., Fig. 5d at ~20 m depth north of 57.4°N), where the vertical diffusion coefficient is below $5 \cdot 10^{-5}$ m$^2$ s$^{-1}$ in a ~20 m deep layer. Applying a typical swimming speed for some dinophytes of ~10 m day$^{-1}$ across a 10 m deep layer in such a region implies a Peclet number >> 1, thus, diel vertical migration is a potential additional nitrogen transport although this cannot be documented from our data.

## 4.5 Nutrient fluxes north of the shelf edge

Nutrient fluxes and NP increased close to the Norwegian coast (Fig. 7, S3). This was in accordance with chlorophyll estimates (Hu et al., 2012) based on daily images from the MODIS-Aqua satellite: A large algal bloom was observed 6 July (not shown) and covered the area off the southern-most part of Norway. A satellite image from 20 July (Fig. 1a) showed increased chlorophyll concentrations near the Norwegian coast and colder sea surface temperatures along the coast also indicated influence from upwelling of subsurface water masses (Fig. 1b). Although our observations were made more than 12 nautical miles from the coast, the increased chlorophyll and PP values at Tr1 and Tr4 are likely related to these features. A tendency to a thicker chlorophyll layer around the SCM and a deeper nutricline at Tr4 and Tr5 also indicates increased production and supply of nutrients near the coast. Mixing processes along the Norwegian shelf edge may, therefore, similarly contribute to NP in the area.

## 5 Conclusions

Relatively high PP and chlorophyll concentrations of 476 mg C m$^{-2}$ d$^{-1}$ and 30 mg chlorophyll m$^{-2}$, respectively, characterised the stratified northeastern North Sea in July 2016. The greatest values were found above the shallow shelf and near the Norwegian coast. Turbulence measurements showed maximum dissipation rates of TKE in the benthic boundary layer above the shallow shelf area and increased mixing above the shelf edge zone, whereas very low vertical mixing rates characterised the deeper open area. Chlorophyll was concentrated in a subsurface chlorophyll maximum located near the nutricline and, on average, about 6 m above the depth of the maximum vertical nitrate flux in the deeper area north of the shelf edge. Chlorophyll was concentrated in the bottom layer above the shallow and nitrate depleted shelf area. The nutricline was located significantly deeper above the shelf edge area (depth ~80 - 130 m) than in deeper water columns. Significant NP was found above the shelf-edge where, at some stations, relatively large nitrate fluxes, i.e. > 0.5 mmol N m$^{-2}$ d$^{-1}$ implied f-ratios above 0.10. In contrast, very low nutrient fluxes characterised the open area above the Norwegian Trench (f-ratios < 0.02). This localised NP along the shelf edge potentially represents an important key to understanding temporal variability in the distribution of organisms (diversity), biological production, and ecosystem structure in this productive area.

**Author contribution**

Measurements of primary production and turbulence were carried out by KR and JB, respectively, and both analysed the data and prepared the manuscript.

**Acknowledgements**

We thank captain and crew on-board R/V Dana for very helpful assistance and support during the cruise and Eik Ehlert Britch for technical support. Erik Askov Mousing carried out most of the PP incubations. This study was supported by funding for ship-time by the Danish Centre for Marine Research. The Carlsberg foundation provided support for the turbulence instrument (CF15-0301). The Villum foundation provided support for the cruise and analysis of the measurements. Analyses were supported by Danish National Science Foundation via its support of the Center for Macroecology, Evolution, and Climate

(grant no. DNRF96). Satellite derived PAR-data was obtained from NASA Goddard Space Flight Center, Ocean Ecology Laboratory, Ocean Biology Processing Group (2014): MODIS-Aqua Ocean Color Data, http://doi.org/10.5067/AQUA/MODIS/L3M/PAR/2018 and http://doi.org/10.5067/AQUA/MODIS/L3M/CHL/2018, (Accessed on 2018/05/10). The manuscript benefitted from comments by two anonymous reviewers.

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

**Figure 1** VERMIX station map of the five transects in the northeastern North Sea overlain on (a) surface chlorophyll a (mg
chl m$^{-3}$) and (b) sea surface temperature (°C) from MODIS satellite images (4 km resolution) obtained 20 July 2016 (clouds are shown in dark gray). (a) CTD stations (bullets) and stations where $^{14}$C-incubations for primary production were used for estimating photosynthetic parameters in the area (white circles) along the five transects (Tr1-5). Horizontal yellow bars (Tr2-5) show the separation between nitrate-deplete and nitrate-replete areas. Water masses in the area are indicated (see text). Bathymetry is contoured in meters.

**Figure 2** Distributions along Tr4 of (a) potential temperature (°C, colours and white contours, additional contours above 16 °C in 0.5 °C intervals), (b) salinity (psu, colours and white contours, additional contours between 26-34 in 0.5 psu intervals) and contours of potential density anomalies (intervals of 1 kg m$^{-3}$, orange lines), (c) oxygen (μmol kg$^{-1}$, colours and black contours) and (c) nitrate (μmol kg$^{-1}$, colours and white contours and orange contours of potential density anomalies as in b). Vertical lines show location of the CTD-measurements.

**Figure 3** Distributions of chlorophyll a (mg chl m$^{-3}$, colours and black contours, additional white contours are shown in intervals of 2 mg chl m$^{-3}$) along Tr1-Tr5 (a-e). Nutricline depths (red bullets) and depths with maximum nitrate flux into the euphotic zone (orange squares) were calculated at each station where water samples (black triangles) and turbulence profiles were made, respectively. Contours of potential density anomalies are shown in intervals of 1 kg m$^{-3}$ (orange lines). Note the different latitude-intervals in the figures.

**Figure 4** (a) Potential temperature-salinity diagram of all CTD-measurements and (b) a high-saline subset of the measurements (small grey bullets). Water sample concentrations of nitrate are shown with large bullets in (a,b) (μmol kg$^{-1}$, colour bar). Contour lines show isopycnals of potential density anomalies and selected water masses are indicated. (c) Nitrate vs phosphate for all water samples. The total water depth is shown with colours (m) and the relationship $[NO_3^-] = \eta_{N:P} [PO_4^-]$, $\eta_{N:P} = 16:1$, is shown with a grey dashed line.

**Figure 5** Turbulence measurements along Tr2 (a,c) and Tr4 (b,d). (a,b) Logarithm (Log$_{10}$) of dissipation of turbulent kinetic energy (W kg$^{-1}$) and (c,d) calculated vertical turbulent diffusion coefficient (m$^2$ s$^{-1}$).

**Figure 6** Vertical profiles from four stations across the shelf edge of (upper panels) potential density anomaly, chlorophyll a and nitrate (bullets) and (lower panels) dissipation of TKE (average value and values from the two shear sensors are shown with bullets and open circles, respectively), Brunt-Väisälä frequency (dashed line), turbulent diffusion coefficient and the

vertical turbulent nitrate flux. The nutricline depth (DNO3) and depth of the euphotic zone (0.1% PAR) are shown (dashed and dotted lines) and station information of locations, $D_{NO3}$ and maximum $F_{NO3}$ to the euphotic zone are shown in the figures (e-h).

**Figure 7** (a) Vertically integrated chlorophyll a (mg chl a $m^{-2}$) and (b) primary production (mg C $m^{-2}$ $d^{-1}$). Values are proportional to the diameter of the circles. (c) Distribution of maximum nitrate flux to the euphotic zone ($F_{NO3}$, mmol N $m^{-2}$ $d^{-1}$) and (e) f-ratios for the euphotic zone (colours, no unit).

**Figure 8** Time series station over 36h at the shelf edge. (a,b) Tidal current speed cubed, ctd-measurements (c, e, g, h) of potential temperature, salinity, oxygen and Brunt Väisälä frequency, respectively. Turbulence measurements of (d) dissipation rate of TKE and (f) calculated vertical turbulent diffusion coefficient. Observations are shown with small gray bullets and samples for water chemistry is shown with bullets in (g)s.

**Figure 9** Sketch of conditions across the shelf edge zone during summer. The largest f-ratios are found above the shelf-edge zone where the nutricline ($D_{NO3}$) gets in contact with increased mixing near the bottom. Deep waters are characterized by high nitrate concentration whereas surface water and water above the shallow North Sea (NS) are nitrate depleted.

**Table 1.** Distribution of photosynthetic parameters. Median values, and absolute median deviations (number of samples *n* in parenthesis) of photosynthetic parameters at 5 m below the surface and at the SCM.

| Depth level | Depth | $P^B_{max}$ * | $P^B_{max}$ | $\alpha^B$ | $\beta^B$ | $E_{max}$ | Chl a |
|---|---|---|---|---|---|---|---|
| | [m] | [µg C (µg chl h)$^{-1}$] | [µg C (µg chl h)$^{-1}$] | $10^{-2}$·[µg C·(µg chl h µE m$^{-2}$ s$^{-1}$)$^{-1}$] | $10^{-3}$·[µg C (µg chl h µE m$^{-2}$ s$^{-1}$)$^{-1}$] | [µE m$^{-2}$ s$^{-1}$] | [mg chl m$^{-3}$] |
| Surface | 5.0± 0.0 (23) | 4.76 ± 1.33 (19) | 5.48 ± 0.87 (23) | 4.10 ± 0.90 (23) | 1.70 ± 1.70 (23) | 413 ± 76 (19) | 0.16 ± 0.06 (23) |
| SCM | 27.0± 5.0 (25) | 1.72 ± 0.38 (24) | 2.33 ± 0.64 (25) | 2.70 ± 0.80 (25) | 3.00 ± 0.90 (25) | 192 ± 26 (24) | 1.67 ± 0.73 (25) |

**Table 2** Median values, and absolute median deviations of all data in three depth intervals, depth of maximum nitrate flux (Depth$_{max}$), maximum nitrate flux into the euphotic zone (FNO3$_{max}$), depth of nutricline (D$_{NO3}$), vertically integrated chlorophyll a (Chl$_{int}$) and primary production (PP).

| Depth interval (m) | Depth (m) | Depth$_{max}$ (m) | FNO3$_{max}$ (mmol N m$^{-2}$ d$^{-1}$) | D$_{NO3}$ (m) | Chl$_{int}$ (mg chl m$^{-2}$) | PP (mg C m$^{-2}$ d$^{-1}$) |
|---|---|---|---|---|---|---|
| 50 - 80 | 65±6 (25) | 39±3 (25) | 0.05±0.04 (25) | 34.5±4.5 (28) | 34.8±13.3 (69) | 476±138 (69) |
| 80 - 130 | 97±11 (18) | 43±3 (18) | 0.11±0.07 (18) | 34.5±3.5 (20) | 26.4±2.9 (24) | 419±41 (24) |
| > 130 | 263±58 (30) | 32±5 (30) | 0.06±0.03 (30) | 26.5±4.0 (31) | 26.6±4.2 (35) | 528±101 (35) |

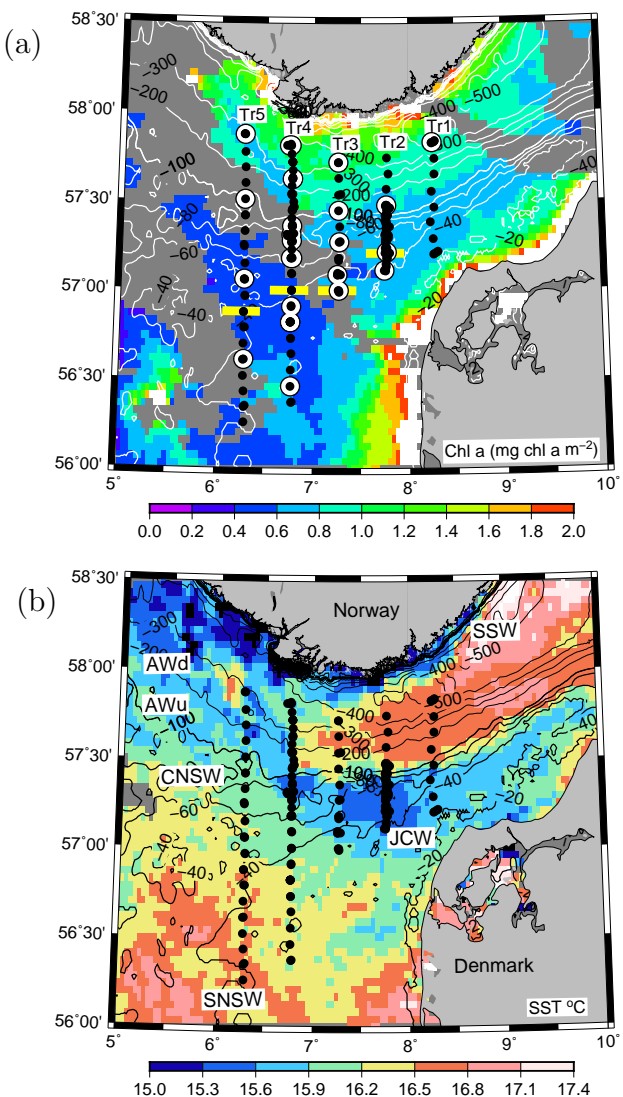

**Figure 1**

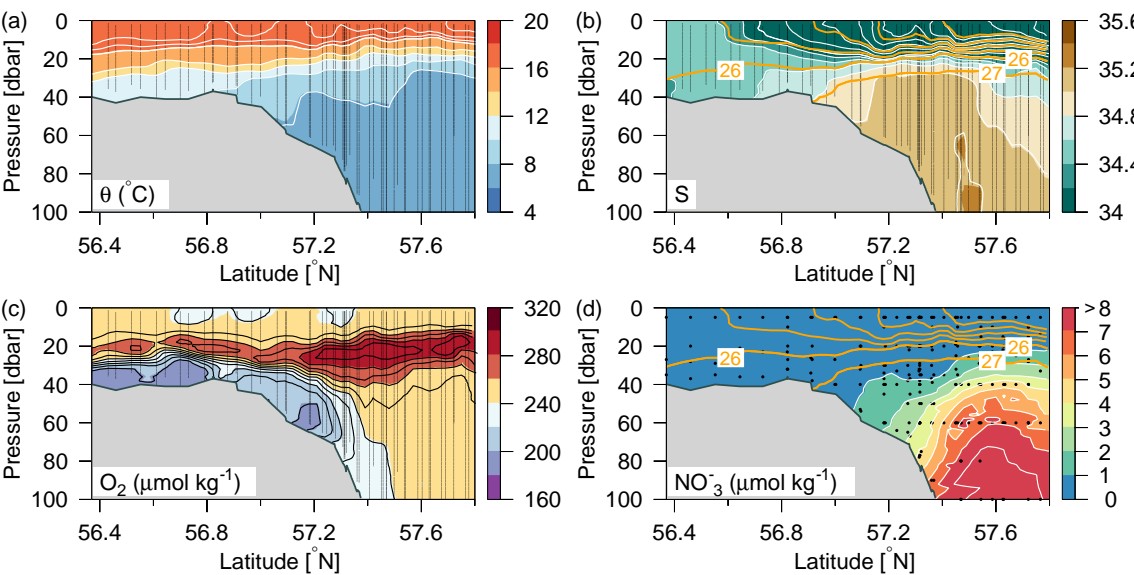

Figure 2

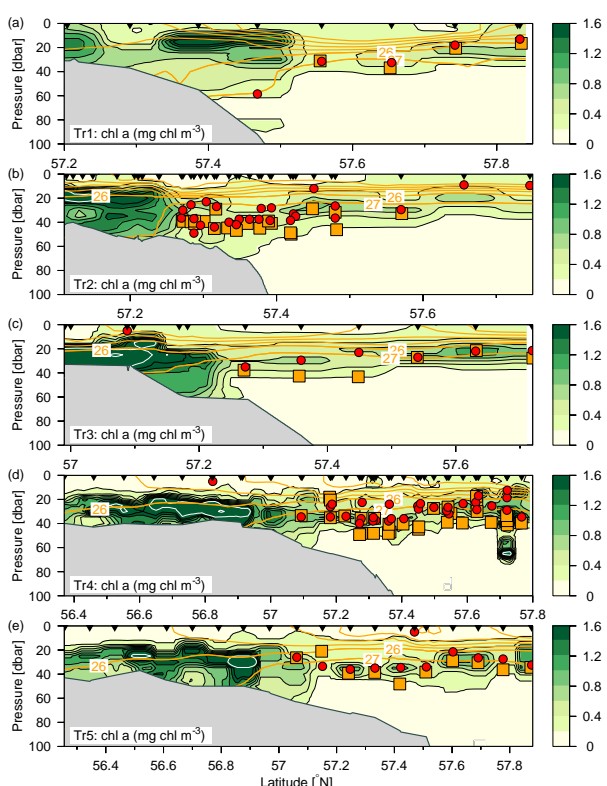

Figure 3

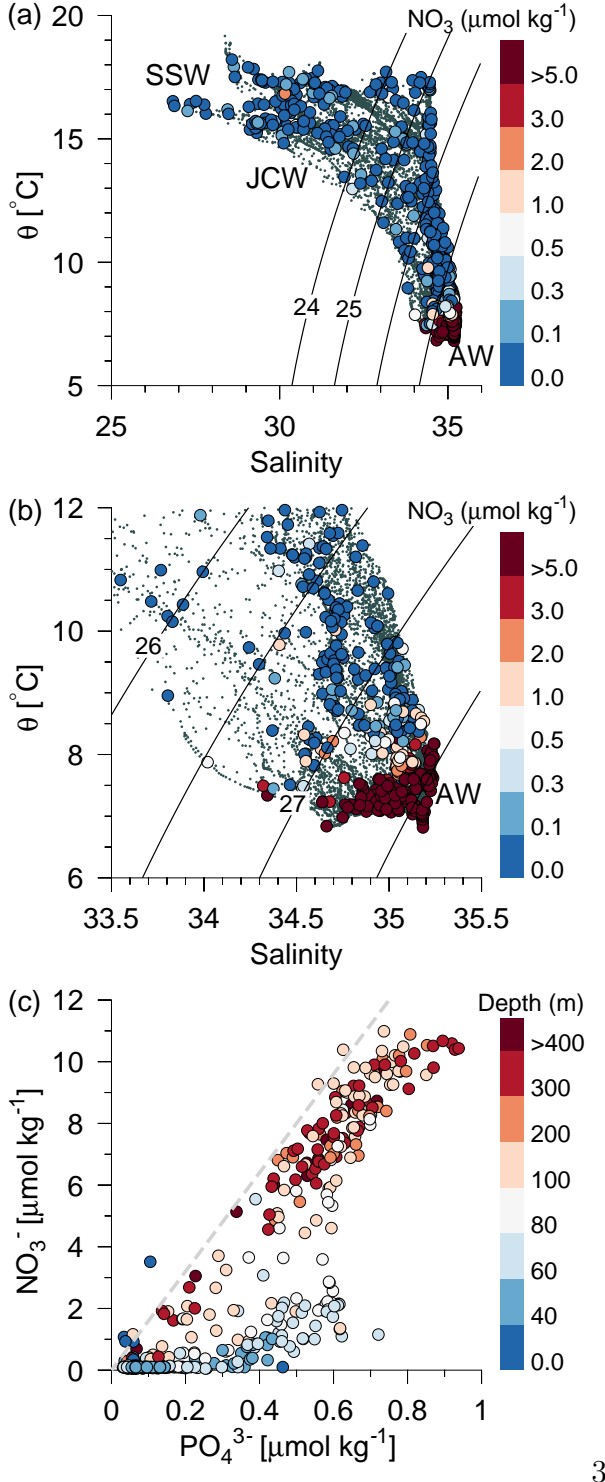

**Figure 4**

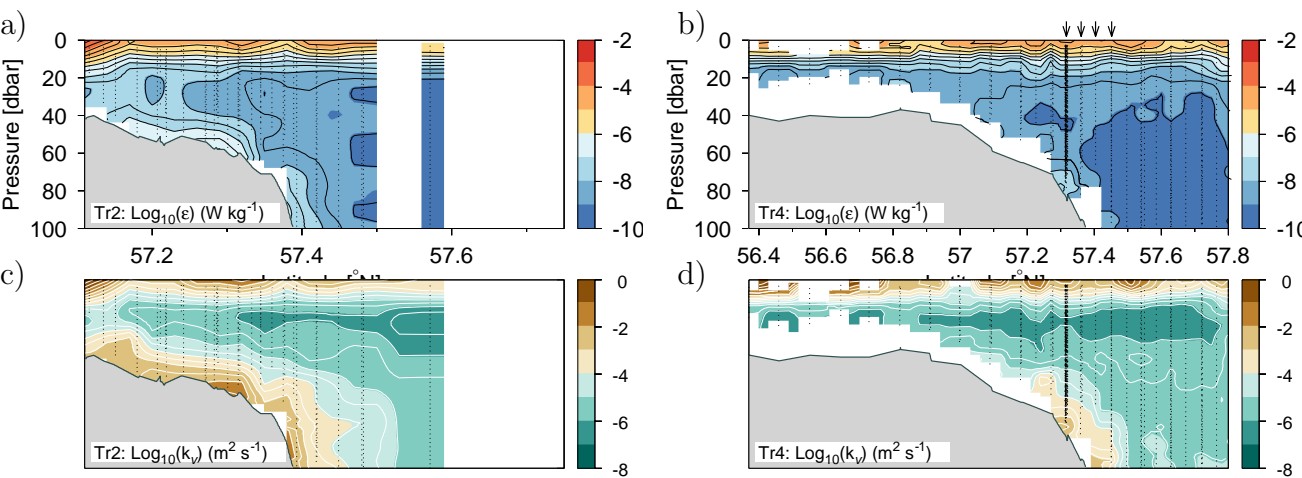

Figure 5

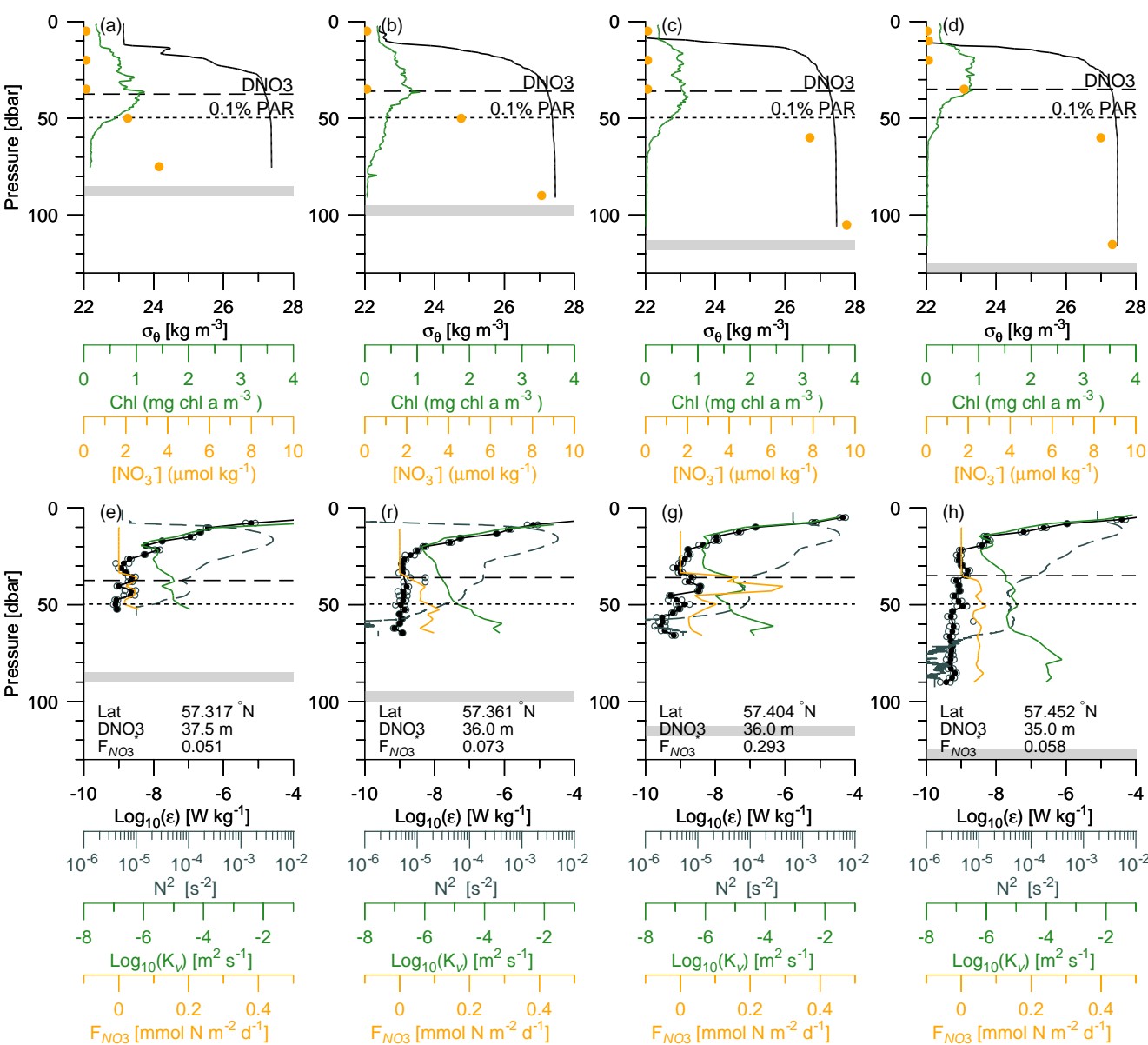

Figure 6

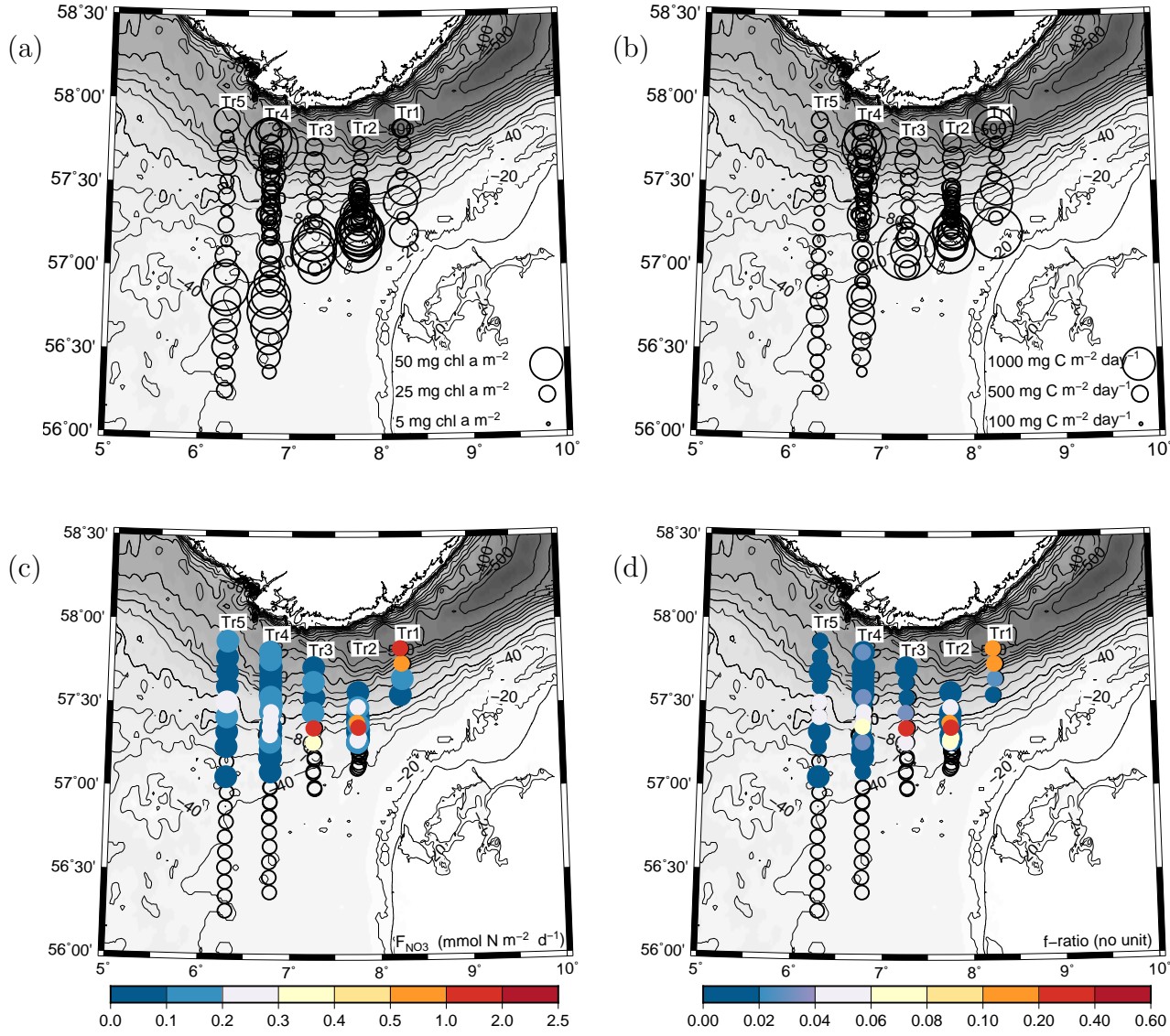

**Figure 7**

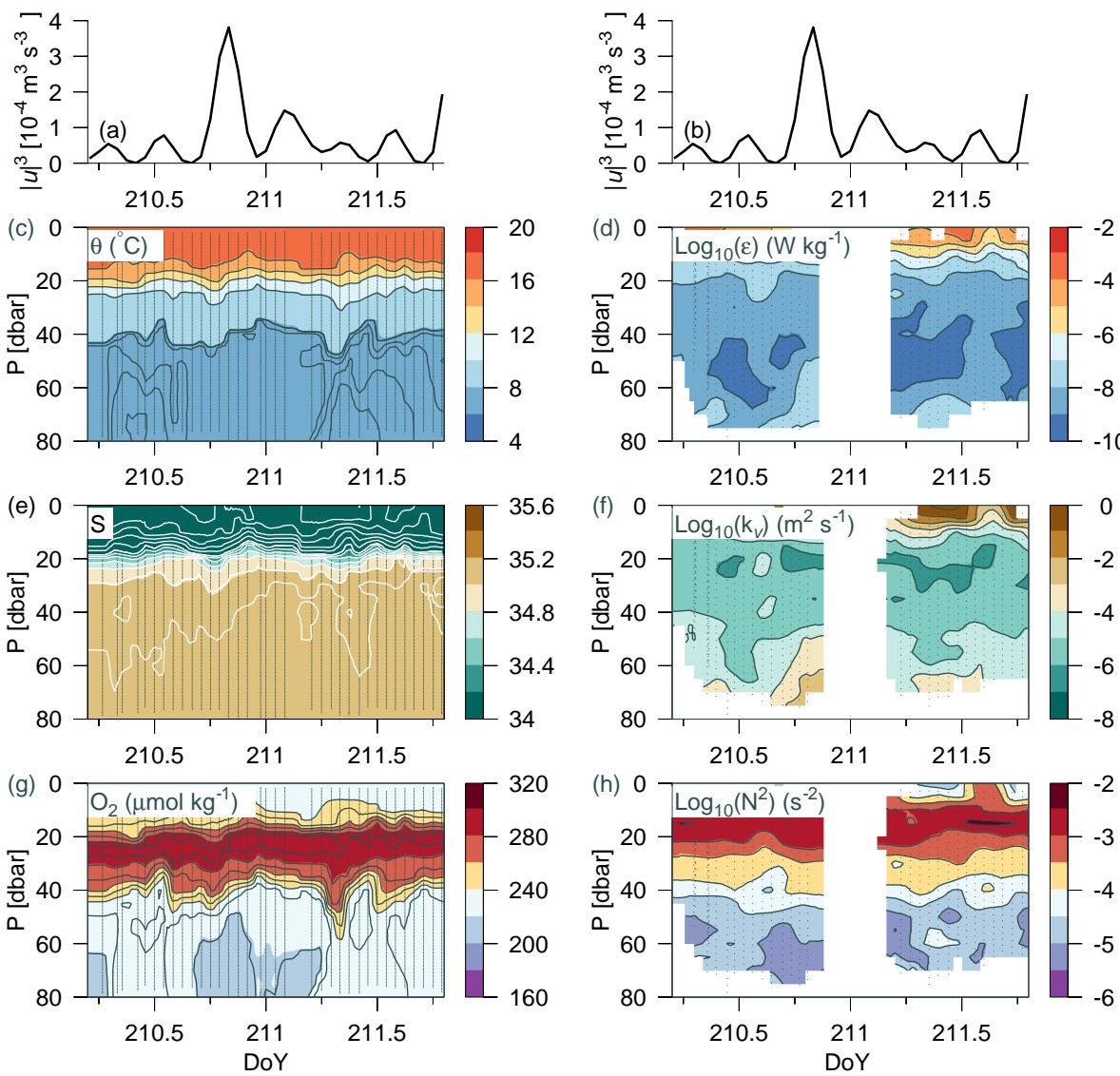

**Figure 8**

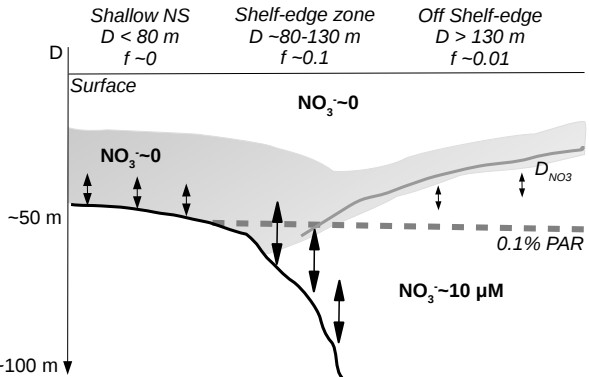

**Figure 9**