# Peer review of "Turbulence measurements suggest high rates of new production over the shelf edge in the northeastern North Sea during summer"

_Biogeosciences, 2018_

## Referee Comment (RC1) · Anonymous Referee #1 · 21 Sep 2018

Review of "Turbulence measurements suggest high rates of new production. . .." By Bendtsen & Richardson.

The paper presents data from an intensive series of stations and transects in the eastern North Sea, reaching from the shelf into the Norwegian trench, to suggest that the edge of the shelf is a site of significantly higher new primary production compared to deeper and shallower regions. The results are certainly interesting, as this shelf edge region is relatively isolated from the open ocean and so is far less influenced by typical shelf edge processes (e.g. internal tides and breaking internal waves). The results

appear to be generally sound, but there is a lack of detail in key areas that needs to be addresses.

General Points:

1. The turbulence data presented is incomplete. Sections on turbulence parameters are presented (Fig. 5), but there is almost no consideration of the typical temporal variability in turbulence. Were the microstructure stations single profiles? Normally turbulence data is collected over a series of profiles to try to capture the chaotic nature of mixing events, and I would expect to see turbulence present with confidence intervals reflecting any variability. Are tidal flows important here? In which case, was there any attempt to provide some average turbulence measurement over a tidal cycle? There is a short statement in the discussion that implies additional data was collected to indicate the amount of temporal variability – if that is the case, it should be included more explicitly in the paper. Also, there is very limited presentation of the nitrate flux data – one profile, and plots of the max flux along transects. A section of the fluxes would be very useful. The paper at times mentions quite strong fluxes below the SCM, which implies a divergence in nitrate flux that needs to be considered.

2. The discussion is somewhat unsatisfactory. Quite a broad range of alternative processes are suggested as underpinning some of the observations, but they are often vague and rather descriptive. Some better quantification of these would help to determine how likely they are as playing important roles. For instance:

(i) On page 12 denitrification is suggested as a mechanism for reducing the shallow water nitrate, with a global mean rate from Yool 2007 mentioned. There are shelf/coastal estimates available, and a quick calculation could be done to assess the feasibility of this process.

(ii) The mechanism for the elevated turbulence at the shelf edge is never discussed. It seems to be a boundary-layer process – is it due to a slope current or tides? Also, the boundary turbulence seems fairly consistent along the transect (e.g. Fig. 5c) –

so is the shelf edge nitrate flux really a result of increased turbulence, or is it because the sloping isopycnals bring the nutricline down towards the turbulence (almost implied on page 15). The latter idea seems to be suggested by Fig. 9 (though without better information on the turbulence data, I'm not convinced that the bed turbulence over the shelf edge is significantly greater than bed-driven turbulence elsewhere – in which case the deepening of the nutricline is vital).

(iii) Isopycnal transport of organic material is suggested as a way of supplying nutrients (page 13), but is not estimated in any way – some reasonable numbers would help in determining its likely use.

(iv) On page 15, "other transport processes" apart from vertical turbulent mixing are required, and motility of phytoplankton is suggested. This again is rather vague – why not quantify the possibility (e.g. use the turbulence data and an estimate of phytoplankton swimming speed to get a Peclet number)?

(v) A link to a coastal bloom off Norway, seen in a MODIS image, is invoked on page 15. Why not show this image, rather than simply assert its likely relevance based on the proximity of the sampling?

Specific Points:

1. Page 2 line 7 (also discussion, page 11 line 29): Linking localised NP to higher trophic levels needs to be more nuanced that implying a simple "more production leads to more fish". Scott et al note that the increased chl arises due to internal wave mixing, and the internal waves might also affect prey aggregation – i.e. the correlation with chl is not causal, but chl and prey aggregation are both a result of internal waves.

2. The introduction/background is very much focused on the North Sea. However, the issues being investigated have much broader significance – it would raise the profile and readership of the paper if a stronger, broader context was provided rather than such a localised one.

3. Page 3, line 18: a 1 km station spacing was used (which is impressive), bit how does that fit alongside the tidal excursion?

4. Page 4, line 10. The mixing efficiency is assumed to be constant, but there's a good deal of recent literature that suggests this is not the case (e.g. Shih et al., J. Fluid Mechanics, 525, 193-214, 2005; Bouffard & Beogman, Dynamics of Atmospheres and Oceans, 61, 14-34, 2013). Both provide a way of estimating efficiency knowing the turbulence intensity – I suspect that the region of data in this N Sea study probably sits where efficiency = 0.2, but it would be good to check this.

5. Page 4, line 16: "the depth of the SCM was sampled" – do this mean the peak of the SCM?

6. Page 4, line 20. Nutrient analyses are mentioned, but no methods – I assume standard methods, but at least cite the usual papers.

7. Page 4 line 25: Why assume that the deep fluorescence signal is not chlorophyll? If you have boundary-driven turbulence acting at the base of the SCM and nutricline then it will draw chl down into the deeper water.

8. Page 5, lines 14-17. I'm not sure why this scaling of observed PAR to the MODIS product was done.

9. Page 6, line 25-26: The assumption of Redfield is a critical part of the results of the paper. Some justification needs to be made to show that the assumption is OK, or to indicate the likely variability of C:N.

10. Page 7, lines 3-5. I struggled to decipher this sentence, please clarify.

11. Page 7, lines 17-20. Re-phrase – this is a very long sentence with inconsistent use of brackets.

12. Page 9, line 14. The highest nitrate flux is reported at a depth below the photic zone. This implies some divergence of the nitrate flux – where does it go if there is no

sink for it?

13. Page 9, lines 26-30. I'm not convinced that the chl-normalised production rates are useful. Chl per cell in the SCM is likely to be higher than in the surface, so comparing chl-normalised parameters does not tell us much. Or does it and I have missed the point? Normalised per cell or per C would make sense (though not clear this is possible).

14. Page 10, line 7: units needed for 4.76 and 1.72.

15. Page 11 line 8. "...coastal upwelling..." This is rather vague. What mechanisms or evidence do you have?

16. Page 14, line 29. The two Sharples refs deal with breaking internal tides/waves. The Burchard & Rippeth ref deals with wind-driven shear spikes and mixing by inertial waves. This is an important aspect of the discussion – most regions of the shelf edge are reported to have high nitrate fluxes due to breaking internal tidal waves. In the present study this is not the case – which is worth pointing out.

17. Page 16, line 1: "...indicate increased mixing, upwelling, or eddy activity..." This is very vague. What evidence do you have, or is there citable work that supports this?

18. Page 20, caption to Fig. 5: (c,d) rather than (b,d).

19. Figs 1 and 7. The bathymetry contours are hard to read. Better labelling needed, also perhaps mark the shelf edge?

20. Figs. 2, 3, 4, 5, 7, the colourbars need units.

21. Fig. 3: parallel sections of density would help a lot in understanding the chl distributions.

22. Fig. 8: the different colours presumably indicate different transects. Legend needed.

---

## Referee Comment (RC2) · Anonymous Referee #2 · 24 Sep 2018

**REVIEW of "Turbulence measurements suggest high rates of new production over the shelf edge in the north-eastern North Sea during summer" by Bendtsen and Richardson**

The manuscript presents an extensive characterization the spatial variability of different variables related to primary and new production across the eastern North Sea shelf. The main conclusion pointed out by the authors is that nitrate turbulent fluxes into the photic layer (ie. new production rates) are enhanced close to the shelf edge, with

potential implications for the ecosystem, as enhanced transfer towards higher trophic levels. The larger turbulent fluxes at the shelf edge do not relate to localized internal wave breaking (as reported for other locations, i.e. the Celtic Sea shelf, Sharples et al. 2007), but to a penetration of the nutricline into the bottom boundary layer following isopycnals, which in turn seem to adjust to the baroclinic flow of Atlantic Water along the shelf edge. The dataset presented is impressive, with a unique collection of biological, chemical and physical parameters, and the results are certainly interesting. The quality of the figures and writing are overall good. However, the manuscript has some significant weak-points that need to be addressed before acceptance. My main comment is that, in my opinion, the results do not convincingly support the main conclusions, at least in the form in which they are presented now (see below).

General comments

1. The main conclusion that the shelf edge is an area of localized nitrate fluxes leading to new production (and increased fishing activity) is not convincingly supported by the results, at least in the way in which they are presented and discussed. If I interpret the text and figures correctly, the integrated values of chlorophyll and primary show a distinct cross-shelf distribution, being minimum close the shelf edge (Page 11, line 6, Figure 7). The authors must explain and discuss why this happens and how this relates to their statement that new production and transfer towards higher trophic levels is enhanced at the shelf edge. I could understand that larger NP may not necessary result in larger PP but this needs to be discussed at least. From figures 7 and 8 it is not entirely clear if f-ratios are larger there because primary production rates are relatively low or because nutrient fluxes are larger. Reporting mean/median values of PP and FNO3 at the different regions (shelf, shelf edge, Norwegian Trench) in Figure 9 would definitely help. Also, an statistical analysis/error assessment would be needed to show that the differences between regions are significant, particularly in the case

of turbulent nitrate fluxes, which are highly uncertain due to the chaotic nature of turbulent mixing. Hence, the presentation and discussion of the results need to be significantly improved. Until then, the title of the manuscript ("Turbulence measurements suggest *high rates of new production* over the shelf edge in the north-eastern North Sea during summer") is not justified.

2. The mechanisms that cause the nutrient fluxes to be larger at the shelf edge are not sufficiently discussed. In particular, it is not clearly shown if larger nitrate fluxes are related to enhanced turbulent dissipation, reduced stratification or enhanced nitrate gradient. I think this is important for the interpretation of the NP dynamics in the area. Additional figures showing the nitrate and buoyancy frequency distribution would help. In the discussion (Section 4.3), the authors point out that the dynamics of the shelf edge in the study area is different from similar locations, where enhanced turbulence and nutrient supply is sustained by internal tide dissipation at the shelf edge (eg. Sharples et al 2007). The authors say that they have carried out some time-series measurements to study the internal wave activity at the shelf edge and they have not found any signal of enhanced mixing (why not show this data at least as Supplementary Information?). They suggest, instead, that the enhanced nutrient fluxes at the shelf edge relate to the deepening of the nitracline at the shelf edge, reaching the bottom boundary layer. This deepening would be related to the baroclinic flow of the nutrient-rich AW at the shelf slope. This could be a very interesting point of the manuscript but it needs to be more clearly demonstrated with data allowing for a more thorough characterization of the site's dynamics, i.e. some current measurements (if available), or at least discussed in more depth with additional support from the literature.

3. Lack of important information: the authors have omitted some relevant information in the methods section and others (see specific comments). Also, at least two figures, which are very relevant for the scientific content of the manuscript,

must be added: (1) the distribution of nitrate concentration along at least one of the transects and (2) a comparison of the modeled vs. measured PP values at *in situ* conditions for the stations where they are available.

4. Structure: The structure of the manuscript is not always linear. I suggest some reorganization of the text/figures (eg. see specific comment 17)

Specific comments

1. Abstract: "Estimated nitrate fluxes due to turbulent vertical mixing into the euphotic zone were up to 0.5 - 1 mmol N m-2 d-1 over the shelf-edge (f-ratios > 0.1) while values of < 0.1 mmol N m-2 d-1 were found in the deeper open area north of the shelf-edge." If this refers to figure 8, those numbers are not easy to read from this figure. A logarithmic scale must be used. Mean/median values (and uncertainties) could be reported in Figure 9.

2. Section 2.2. Important information is lacking in this section. What was the final vertical resolution of the TKE dissipation rate? How many casts were performed at each station?

3. Section 2.3. How many nutrient and chlorophyll profiles/samples were analysed? "In some cases/At some stations" are very vague expressions. What was the intended horizontal and vertical resolution for nutrients? How were the sampling stations chosen?

4. Section 2.4, Page 5, lines 11-12. The goodness of the fits to eq. (3) is not sufficiently demonstrated. The authors should provide any measurement of this goodness and/or some plot of the data and fitted lines.

5. What is the difference between PBmax and PBmax*?

6. FNO3 calculation. If I understood correctly, the FNO3 fluxes into the photic zone at each station are reported as the maximum of the FNO3 across the nitracline. Thus, the reported fluxes are the result of a point by point multiplication of "measured" Kv values and calculated NO3 gradient. Kv has generally a patchy distribution in space and episodic in time, so that the fluxes calculated in this way my contain spurious values. How did the authors deal with this? Did they apply any averaging to the "measured" Kv values? How many casts were done at each station? The robustness of the FNO3 calculation must be assessed through a more thorough error analysis.

7. Page 7, lines 7-23, Fig. 2. The authors could identify the different water masses with a text label in Figure 2. Also, the authors may outline the main circulation patterns of the different water masses in Figure 1 and provide some geographic indications (name of the countries and some topographic) features to facilitate the orientation of the reader.

8. Page 7, lines 7-23, Page 8 lines 1-23. Though extremely relevant for the study and extensively described in these lines, nitrate distributions are not shown in the manuscript. The authors must at least include the nitrate distribution of transect 4 in Figure 2.

9. Page 8. I don't believe that adding a new subsection (3.2.1) is necessary here.

10. Fig. 3: there is some overlapping between the red circles and orange squares and in some cases it is difficult to know whether some points are lacking or hidden . You could use different sizes

11. Figs. 2. and 4. In the methods section, the authors say that sections 2 and 4 were repeated to study the temporal variability. Are the distributions presented in Figs. 2 and 4 a mean of the different occupations, or how were they calculated?

12. Fig. 6. The vertical distribution of FNO3 is very difficult to appreciate in this figure because it follows the logarithmic variability of Kv. The authors may use a log-scale for FNO3 too (also in Fig. 8). The largest FNO3 are shown for the lower boundary layer, due to larger values of the diapycnal diffusion coefficient. The nitrate gradient however is very weak here, so I doubt whether these large fluxes would actually different from zero if the uncertainties in the nitrate gradient calculation and Kv were accounted for. Error bars should be added to the nitrate flux.

13. Section 3.3. I would have expected to find a description of the spatial distribution of the nitrate fluxes here similar to previous sections.

14. Section 3.5 / Figure 7. Vertically integrated quantities (Chlorophyll and PP) are reported in this section/figure. However, I could not find the integration depth in the manuscript. I guess that they have been integrated in the euphotic zone but this should be specified.

15. Section 3.5, Page 10 Line 19. How do the extrapolation with equation 3 compares with measured PP at local conditions at the locations where direct measurements are available? I suggest to add a new figure where modeled and measured values are compared.

16. Figure 7c and text. There is some overlapping of the color dots here and it is difficult to see whether there is a clear background tendency towards higher f-ratios at the shelf edge or there are only a few large values superimposed to a generally low background. How does this relate to the episodic nature of turbulent mixing? I would suggest to calculate average f-ratios for the shelf, the shelf-edge and the Norwegian Trench based on the mean (or median values) of PP and FNO3 in the different regions, instead of the point-wise calculation presented here. This numbers could be shown in Figure 9. This would also allow for a

quantitative evaluation of the significance of the differences in NP between the different areas.

17. Sections 3.5 an 3.6 / Figures 7 and 8: The information about the spatial distribution of PP and integrated chlorophyll-a is somehow dispersed and repeated in these two figures/sections. On the other hand, in my opinion, the description of the spatial variability of the nitrate fluxes -which seems to be a central topic of the manuscript- is insufficient. I would replace the f-ratio in Fig. 7 by the actual nitrate flux and describe its variability and drivers (changes in nitrate gradient, stratification and TKE dissipation) in section 3.3, for example.

18. Figure 8. The location of the shelf edge is not evident at all in this figure and this weakens the authors' main point (new production is enhanced at the shelf edge). I would suggest to represent the different variables as a function of the distance to the shelf edge instead of latitude. The smooth cross shelf distribution of FNO3 and the f-ratio outlined in Figure 9 and the abstract (see first comment) is not clear in this figure due to the large short-scale variability of these quantities. I would suggest to use logarithmic scale or even add a representation of FNO3 in figure 7, report mean values in Figure 9, and remove figure 8.

19. Figure 9. This figure is promising but it definitely needs more information. I would add mean values of primary production and nitrate fluxes (at least). From Figures 7 and 8 it is very difficult to know if the larger f-ratios at the shelf edge are mostly due to enhanced nitrate fluxes or reduced primary production in this area. How were the f-ratios calculated, are they mean/median values or just an estimate of their order of magnitude? This is the main message of the manuscript and the authors should provide a solid quantification (and some error assessment) of the f-ratio.

20. Section 4.1. This section could be much improved if a comparison between modeled and measured PP values was shown.

21. Page 14, lines 10. "Finally, estimates of new production imply a conversion from nitrate to carbon and a fixed ratio may not be representative for the different communities in the area." Is not there any quantification of plankton stoichiometry in the area available to assess the validity of the chosen C:N ratio?

22. Page 14, line 29-30: "Mixing from tides (Sharples et al, 2007; 2009) and breaking internal waves (e.g. Burchard and Rippeth, 2008) has been shown to be important for vertical nutrient fluxes in shelf areas." This sentence is imprecise. In Sharples et al. (2007) mixing is enhanced due to internal wave breaking (in particular to the dissipation of the internal tide) and in Burchard and Rippeth (2009) enhanced turbulence is due to the the alignment of the shear vectors induced by different sources (inertial oscillations, wind and tidal bed friction). Also, the Burchard paper is from 2009, not 2008. In general this section has great potential, but needs to be improved (see General comment 2)

23. Page 15, Lines 18-26. This paragraph does not match the section heading

Technical comments

1. Page 6, line 21 and Page 9 line 10. There are too much ")"

2. Page 17, line 7. Rippith → Rippeth

---

## Author Comment (AC1) · 29 Oct 2018

Reviewer 1: The paper presents data from an intensive series of stations and transects in the eastern North Sea, reaching from the shelf into the Norwegian trench, to suggest that the edge of the shelf is a site of significantly higher new primary production compared to deeper and shallower regions. The results are certainly interesting, as this shelf edge region is relatively isolated from the open ocean and so is far less

influenced by typical shelf edge processes (e.g. internal tides and breaking internal waves). The results appear to be generally sound, but there is a lack of detail in key areas that needs to be addresses.

General Points: 1. The turbulence data presented is incomplete. Sections on turbulence parameters are presented (Fig. 5), but there is almost no consideration of the typical temporal variability in turbulence. Were the microstructure stations single profiles? Normally turbulence data is collected over a series of profiles to try to capture the chaotic nature of mixing events, and I would expect to see turbulence present with confidence intervals reflecting any variability. Are tidal flows important here? In which case, was there any attempt to provide some average turbulence measurement over a tidal cycle? There is a short statement in the discussion that implies additional data was collected to indicate the amount of temporal variability – if that is the case, it should be included more explicitly in the paper. Also, there is very limited presentation of the nitrate flux data – one profile, and plots of the max flux along transects. A section of the fluxes would be very useful. The paper at times mentions quite strong fluxes below the SCM, which implies a divergence in nitrate flux that needs to be considered.

Response: We would like to thank the reviewer for a careful positive review with constructive comments and criticism of our manuscript. The reviewer's concerns put in focus several important issues that we now address in more detail in the revised manuscript. Specifically, we have added more information from the two time series stations (T1, T2) which were only briefly described in the original version of the manuscript. We use time series data from T1 (and further details in Supplementary information and figure S1) to estimate the significance of the epsilon-estimates and for analyzing the temporal variability at the shelf edge. The time series data set includes 107 profiles made in three sequences at one site over a 22-hour period with a time interval of about 3 minutes. With this data set, we analyse the variability between the two simultaneous measurements from the two shear-sensors and we estimate the uncertainty associated with the differences between the two shear probe measurements.

In the Supplementary Information we show that the differences are in qualitative accordance with a normal distribution characterized by the absolute deviation of the samples. This allows us to apply a quality criterion on the data included in the study, i.e., that the differences between sensor readings should be less than three times the absolute deviation. We apply this criterion to all of the measurements used in the study. A very limited number of measurements did not meet the criterion described above. This supports the validity of our approach of using the average value of the two shear probes from a single profile as providing a representative turbulence value for the stations visited on our transects. The relatively close station spacing on our transects as well as the spatial distribution also indicates a consistent distribution pattern for vertical mixing parameters in the area. In addition, we use the time series data from T1 to consider the temporal variation between subsequent profiles separated by only 3 minutes. These new analyses are described in the text in the revised manuscript and the details provided in the revised Supplementary Information section and in Supplementary figure 1.

Further information on spatial variability is also now included in figure 6, where the measurements from four stations, separated by 5 km, and made within four hours are presented. In addition, we have added a new figure 8, as suggested by the reviewer, where data from the second time series station are analysed (this time series station was previously only described in the text). We also analyse temporal variability in water column characteristics in relation to tides and other energy sources. This analysis shows that increased mixing in the boundary layer is in phase with tidal energy input but also that energy from non-tidal currents may be important in this area. For example, a short-term change of T, S and O2 can be associated with advection of ambient water masses, i.e. not directly related to tidal flow.

Nitrate fluxes are now shown for four stations in figure 6 and the spatial distribution of the maximum nitrate flux is also now added to figure 7. Finally, average values for nitrate concentration in three depth intervals are presented in the new Table 2.

Reviewer 1: 2. The discussion is somewhat unsatisfactory. Quite a broad range of alternative processes are suggested as underpinning some of the observations, but they are often vague and rather descriptive. Some better quantification of these would help to determine how likely they are as playing important roles.

Response: We believe that explaining the observed distributions of, for example, nitrate, in our study requires knowledge of seasonal nutrient dynamics in this area that we do not have. Therefore, our considerations of nutrient distributions are to some extent qualitative and we refer to previous studies to provide background information on these aspects. Nevertheless, we have now considered the specific points raised by the reviewer below and clarified the considerations in the Discussion section.

Reviewer 1: For instance: (i) On page 12 denitrification is suggested as a mechanism for reducing the shallow water nitrate, with a global mean rate from Yool 2007 mentioned. There are shelf/coastal estimates available, and a quick calculation could be done to assess the feasibility of this process.

Response: We originally referred to the global nitrification rate of Yool et al. (2007) to explain the potential for recycling of ammonium in the water column. In this revised version of the manuscript, we have also included a reference from Fan et al. (2015) on denitrification rates and show that the apparent loss of nitrate recorded here could be explained by these rates.

Reviewer 1: (ii) The mechanism for the elevated turbulence at the shelf edge is never discussed. It seems to be a boundary-layer process – is it due to a slope current or tides? Also, the boundary turbulence seems fairly consistent along the transect (e.g. Fig. 5c) – so is the shelf edge nitrate flux really a result of increased turbulence, or is it because the sloping isopycnals bring the nutricline down towards the turbulence (almost implied on page 15). The latter idea seems to be suggested by Fig. 9 (though without better information on the turbulence data, I'm not convinced that the bed turbulence over the shelf edge is significantly greater than bed-driven turbulence elsewhere

[Figure]

– in which case the deepening of the nutricline is vital).

Response: The reviewer's points here are in accordance with our understanding of the processes at the shelf edge. We have added more material to clarify and support this point. A figure has been added (Fig. 8) showing the temporal variability at the shelf edge where mixing in the bottom boundary layer is seen to increase and elevated mixing in periods reaches the bottom of the euphotic zone. Thus, interference between a deep nutricline and bottom mixing may provide a mechanism for enhancing diapycnal nitrate fluxes. We have also added Table 2 in which it is shown that the nutricline is significantly deeper above the shelf edge than in the deeper areas. This is also seen in figure 2d (where we have added the nitrate concentration along Tr4) and shown in figure 3 (where we have added isopycnals to illustrate the link between the nutricline, chlorophyll and the density fields) and in the conceptual figure 9. Thus, the deepening of the nutricline, together with increased mixing in the bottom boundary layer, is probably an important mechanism for the elevated nutrient fluxes above the shelf edge. We speculate on this in the Discussion, in particular in relation to figure 9 where the potential dynamic feedback between currents along the shelf-edge, the depth of the nutricline and nutrient fluxes into the euphotic zone is outlined. Thus, this proposed mechanism is, indeed, a result of the deepening of the nutricline and elevated mixing above the shelf edge area.

Reviewer 1: (iii) Isopycnal transport of organic material is suggested as a way of supplying nutrients (page 13), but is not estimated in any way – some reasonable numbers would help in determining its likely use.

Response: It is difficult to provide better information than the time scales for the decay of organic matter we describe in the text. The relevant time scales are of the order days to weeks and we refer to a previous study where we analysed these time scales; Thus, even small cross-shelf transports may contribute with isopycnal fluxes. However, we do not have any measurements of the labile fraction of organic matter in the area. In addition, we have only limited information on the cross-shelf exchange. So, rather

than speculating further on this issue, we prefer to relate to the information on the time scales described in the text.

Reviewer 1: (iv) On page 15, "other transport processes" apart from vertical turbulent mixing are required, and motility of phytoplankton is suggested. This again is rather vague – why not quantify the possibility (e.g. use the turbulence data and an estimate of phytoplankton swimming speed to get a Peclet number)?

Response: We have now added an example at the end of Section 4.4, based on swimming speeds of dinoflagellates (Raven and Richardson,1984), to illustrate the potential of diel vertical migration to provide access to nutrients. In addition, we now also estimate the associated Peclet number in the area north of the shelf edge where vertical diffusion coefficients are very low and show that Pe » 1.

Reviewer 1: (v) A link to a coastal bloom off Norway, seen in a MODIS image, is invoked on page 15. Why not show this image, rather than simply assert its likely relevance based on the proximity of the sampling?

Response: We have followed this suggestion and added the MODIS-derived fields of chlorophyll a and SST in figures 1a and 1b, respectively. We refer to the satellite images in the discussion.

Reviewer 1: Specific Points: 1. Page 2 line 7 (also discussion, page 11 line 29): Linking localised NP to higher trophic levels needs to be more nuanced that implying a simple "more production leads to more fish". Scott et al note that the increased chl arises due to internal wave mixing, and the internal waves might also affect prey aggregation – i.e. the correlation with chl is not causal, but chl and prey aggregation are both a result of internal waves.

Response: We have clarified the paragraph. We agree with the view that Scott et al related increased NP to increased mixing, and this was also the intention with the paragraph.

Reviewer 1: 2. The introduction/background is very much focused on the North Sea. However, the issues being investigated have much broader significance – it would raise the profile and readership of the paper if a stronger, broader context was provided rather than such a localised one.

Response: We have followed the suggestion by the reviewer and added a paragraph in the end of the Introduction where we relate to the more general implications of shelf-edge processes and to conditions in similar shelf-regions.

Reviewer 1: 3. Page 3, line 18: a 1 km station spacing was used (which is impressive), bit how does that fit alongside the tidal excursion?

Response: A rough estimate of the tidal excursion, based on SST-evolution in forecast models of the North Sea, is ∼5 km, and this has to be considered when samples from closely spaced stations are analyzed. The station spacing was gradually decreased along a section of Tr2 for analyzing sub-mesoscale changes in plankton communities. This aspect is not the focus of this study and, therefore, not discussed further in this manuscript.

Reviewer 1: 4. Page 4, line 10. The mixing efficiency is assumed to be constant, but there's a good deal of recent literature that suggests this is not the case (e.g. Shih et al., J. Fluid Mechanics, 525, 193-214, 2005; Bouffard & Beogman, Dynamics of Atmospheres and Oceans, 61, 14-34, 2013). Both provide a way of estimating efficiency knowing the turbulence intensity – I suspect that the region of data in this N Sea study probably sits where efficiency = 0.2, but it would be good to check this.

Response: The reviewer's comments have now been considered. We have added a paragraph in the Methods section where we show that the range where a constant mixing efficiency of 0.2 is valid encompasses the values we apply in the calculation of the nutrient fluxes into the euphotic zone. We have also added the two references brought to our attention by the reviewer.

Reviewer 1: 5. Page 4, line 16: "the depth of the SCM was sampled" – do this mean the peak of the SCM?

Response: Yes, and we have added a comment to clarify this.

Reviewer 1: 6. Page 4, line 20. Nutrient analyses are mentioned, but no methods – I assume standard methods, but at least cite the usual papers.

Response: We have added more information on the nutrient analysis and included a reference to Grasshoff et al. (1983).

Reviewer 1: 7. Page 4 line 25: Why assume that the deep fluorescence signal is not chlorophyll? If you have boundary-driven turbulence acting at the base of the SCM and nutricline then it will draw chl down into the deeper water.

Response: The background value was determined from a deep station (522 m) where a relatively constant fluorescence value was observed between 100 m and 500 m. We see no reason to assume that chlorophyll would be uniformly distributed throughout this deep layer. Therefore, we treated this relatively small background fluorescence as being derived from an unknown source and it was subtracted from the fluorescence signal before the calibration. We have reformulated the sentence to clarify this.

Reviewer 1: 8. Page 5, lines 14-17. I'm not sure why this scaling of observed PAR to the MODIS product was done.

Response: We have added "during the day" to clarify this. The integral in Eq. 3 includes the daily variation of the insolation and this influences the integrated primary production significantly because of the non-linear terms in the equation. It has also been clarified by adding the time and depth dependence, i.e. (t,z), of the variables in the integral.

Reviewer 1: 9. Page 6, line 25-26: The assumption of Redfield is a critical part of the results of the paper. Some justification needs to be made to show that the assumption is OK, or to indicate the likely variability of C:N.

Response: The Redfield ratio is characterized by a C:N ratio of 106:16, and this ratio is widely used in observational and model studies, although variation of the ratio is known to occur. Thus, applying a constant ratio introduces an additional error-source in the calculations. We have added the original reference of Redfield et al. (1963) where a general relationship between the elemental stoichiometry of C:N:P in plankton is documented.

Reviewer 1: 10. Page 7, lines 3-5. I struggled to decipher this sentence, please clarify.

Response: We have clarified the introductory sentences.

Reviewer 1: 11. Page 7, lines 17-20. Re-phrase – this is a very long sentence with inconsistent use of brackets.

Response: We have reformulated the sentence.

Reviewer 1: 12. Page 9, line 14. The highest nitrate flux is reported at a depth below the photic zone. This implies some divergence of the nitrate flux – where does it go if there is no sink for it?

Response: To answer this question, we would need more measurements from the area around the station and in the boundary layer. There is a temporal change, likely associated with the tidal currents, as shown in the new figure 8. Figure 6 in the original manuscript has now been replaced with a section showing more profiles taken across the shelf edge at Tr4 obtained over a short period. However, the divergence is likely associated with transient currents and an example of this is now shown and discussed in relation to the time series station in the new figure 8.

Reviewer 1: 13. Page 9, lines 26-30. I'm not convinced that the chl-normalised production rates are useful. Chl per cell in the SCM is likely to be higher than in the surface, so comparing chl-normalised parameters does not tell us much. Or does it and I have missed the point? Normalised per cell or per C would make sense (though not clear this is possible).

Response: Primary production is, as here, frequently calculated from the model of Platt et al. (1980) in Eq. 3. We have clarified this procedure by adding the integral over the vertical depth range (from the bottom of the euphotic zone to the surface) and during the day (24h, i.e. only daytime PAR-values contribute). Thus, the integral considers the vertical chlorophyll a concentration (we now specify the depth-and time-dependence of the parameters in the integral) and, therefore, the PBmax-values are normalised in the equation. We sample data from both the surface and the SCM (cf. Table 1) to take the potential vertical variation of the phytoplankton characteristics, as mentioned by the reviewer, into account - both in relation to photosynthetic parameters and chlorophyll content. The normalisation implies that when PBmax(z) is multiplied with the chlorophyll concentration in Eq. (3) then this variation is accounted for.

Reviewer 1: 14. Page 10, line 7: units needed for 4.76 and 1.72. Response: The unit has been added.

Reviewer 1: 15. Page 11 line 8. ". . .coastal upwelling. . ." This is rather vague. What mechanisms or evidence do you have?

Response: In addition to our measurements showing increased vertical fluxes, satellite images also indicate that coastal upwelling may be significant along the Norwegian coast. We now refer to the new figure 1b, showing SST from a MODIS-image and we have added the following to the text: ("also indicated by relatively cold Norwegian coastal water masses observed from satellite in Fig. 1b").

Reviewer 1: 16. Page 14, line 29. The two Sharples refs deal with breaking internal tides/waves. The Burchard & Rippeth ref deals with wind-driven shear spikes and mixing by inertial waves. This is an important aspect of the discussion – most regions of the shelf edge are reported to have high nitrate fluxes due to breaking internal tidal waves. In the present study this is not the case – which is worth pointing out.

Response: We have corrected the description of the references. From our data, we cannot identify the specific processes behind the mixing, and this is now clarified in the

paragraph.

Reviewer 1: 17. Page 16, line 1: ". . .indicate increased mixing, upwelling, or eddy activity. . ." This is very vague. What evidence do you have, or is there citable work that supports this?

Reponse: We have reformulated the sentence to: "A tendency towards a thicker chlorophyll layer around the SCM and a deeper nutricline at Tr4 and Tr5 also indicates increased production and supply of nutrients near the coast." We now refer to the upwelling elsewhere in the text where we refer to the SST seen in the new figure 1b. Therefore, this is not repeated in this section.

Reviewer 1: 18. Page 20, caption to Fig. 5: (c,d) rather than (b,d).

Response: This has been corrected.

Reviewer 1: 19. Figs 1 and 7. The bathymetry contours are hard to read. Better labelling needed, also perhaps mark the shelf edge?

Response: The font size has been increased in the figures so it is easier to read the depth contours (it should now be easier to identify the shelf edge so no additional lines are included).

Reviewer 1: 20. Figs. 2, 3, 4, 5, 7, the colourbars need units.

Response: The units of the color bar have now been added to the figures and described in the figure legends.

Reviewer 1: 21. Fig. 3: parallel sections of density would help a lot in understanding the chl distributions.

Response: Contour lines of density have been added, as suggested, to all the panels in figure 3.

Reviewer 1: 22. Fig. 8: the different colours presumably indicate different transects.
Legend needed.

Response: Line legends for all the panels are shown in panel (f) and this information is now also added to the figure legend.

Please also note the supplement to this comment:
https://www.biogeosciences-discuss.net/bg-2018-385/bg-2018-385-AC1-supplement.pdf

**Supplement:**

[revised manuscript text omitted]

Figure 2

[Figure]

Figure 6

[Figure]

**Figure 7**

33

[Figure]

**Figure 8**

**Supplementary material**

**Analysis of turbulence variation and uncertainty**

Temporal variation and uncertainty associated with the shear probe measurements were assessed at a time series station, T1, located on Tr2 (57.287 °N, 7.758 °E; 62 m deep) starting 20 July 23:25 and ending 21 July 21:44 (Fig. S1). In tota,l 107 profiles were made to ~50 m depth in three sequences during the period with typically ~3 minutes between each cast. The temporal variation showed a modest change in temperature between 10-20 m (Fig. S1a) and a relatively small dissipation of turbulent kinetic energy (TKE) at mid-depth during the first 6 hours of the measurements (Fig. S1b). The instrument was equipped with two shear probes and the dissipation of TKE ($\varepsilon$) was calculated from each of the probes. The difference between these two estimates, made across the same water volume, are analysed below to assess the uncertainty of the $\varepsilon$-values.

Samples above 15 m were disregarded in the error-analysis to avoid any influence from the movement of the ship. The relative difference between the calculated dissipation of TKE ($\varepsilon$) obtained from each of the two shear probes (i.e., $\varepsilon_1$ and $\varepsilon_2$) was calculated as: $\Delta Log_{10}\varepsilon = Log_{10}(\varepsilon_1) - Log_{10}(\varepsilon_2)$. In total, there were 1145 pairs over the 22h period with a relatively small $\Delta Log_{10}\varepsilon$ average value of -0.063 and a standard and absolute deviation of 0.23 and 0.14, respectively. We applied the absolute deviation, i.e. the more conservative estimate, as being representative of the uncertainty of the $\varepsilon$-values. The relative probability distribution of $\Delta Log_{10}\varepsilon$ showed a qualitative accordance with a normal distribution characterised by the average value and absolute deviation (Fig. S1b), although the error-distribution showed a tendency to a broader variance for $\Delta Log_{10}\varepsilon$ larger than ~0.4. This was also clear from the cumulative probability distribution (Fig. S1d) where the error-distribution deviated from a normal distribution (confirmed by a Kolmogorov-Smirnov test). We considered the largest values of $\Delta Log_{10}\varepsilon$ to indicate sources of errors which could not be directly related to the instrument but potentially associated with the measurement procedure, for example influence from the rope on the free-falling instrument (all casts were made with free and undisturbed line to the free falling instrument during the whole cast). Therefore, we applied the criterion that only measurements where $\Delta Log_{10}\varepsilon$ was less than 3 times the absolute deviation (i.e. 0.42) were considered to be acceptable and these were included in the analysis. This criterion eliminated only a small number of the $\varepsilon$-values from the data set.

Temporal variation was also considered from the time series measurements at T1. Variation of $\varepsilon$ is expected to vary due to tides, wind, breaking internal waves etc. Therefore, variations at a single time series station cannot be expected to be representative for the data set as a whole. However, the short-term temporal variation was analysed from samples of $\varepsilon$ binned in 5 m intervals and analysed over a period of 30 minutes (i.e. 11 casts) resulting in average values and absolute standard deviations of 1.6±0.6, 1.4±0.6 and 2.1±1,.0 in depth intervals between 25-30m, 30-35m and 35-40m, respectively (in units of $10^{-9}$ W kg$^{-1}$). Thus, short term variation was relatively small and temporal changes between subsequent casts were considered to have a small influence on the calculated $\varepsilon$-values. Therefore, $\varepsilon$-values were, in general, derived from a single cast between the relatively closely spaced stations, where the $\varepsilon$-value obtained by averaging the calculated value from the two probes was reported. In addition to the time series station, T1, a similar time series station (T2) located at Tr4 is discussed in the text.

**Supplementary figure legends**

**Figure S1** Turbulence measurements from time series station, T1, located on Tr2 (57.287 °N, 7.758 °E) starting 20 July 23:25 and ending 21 July 21:44. (a) Temperature (°C) and (c) turbulent kinetic energy dissipation ($\varepsilon$, W kg$^{-1}$) were measured in 107 profiles (small bullets) in three sequences during the 22h period and are shown as a function of pressure and time (Day of the Year). (b) The error-distribution ($\Delta \text{Log}_{10}\varepsilon$ in units of $\text{Log}_{10}$(W kg$^{-1}$), see text) between calculated $\varepsilon$ from the two shear probes (gray bars) and the normal distribution (green) associated with the average and absolute deviation of the error-distribution. (d) The cumulative probability of the error-distribution (black) compared with the associated normal distribution (green).

**Figure S2** Distributions along the five transects of (a) vertically integrated chlorophyll a (mg chl a m$^{-2}$), (b) primary production (mg C m$^{-2}$ d$^{-1}$), (c) nutricline depth (m), (d) maximum nitrate flux into the euphotic zone (mmol N m$^{-2}$ d$^{-1}$) and (e) f-ratio in euphotic zone. Repeated stations on Tr2 and Tr4, separated in time by about a week, are shown with bullets and open circles.

[Figure]

Figure S1

29

---

## Author Comment (AC2) · 29 Oct 2018

Anonymous Referee #2 Reviewer 2: The manuscript presents an extensive characterization the spatial variability of different variables related to primary and new production across the eastern North Sea shelf. The main conclusion pointed out by the authors is that nitrate turbulent fluxes into the photic layer (ie. new production rates) are enhanced close to the shelf edge, with potential implications for the ecosystem, as enhanced transfer towards higher trophic levels. The larger turbulent fluxes at the shelf edge do not relate to localized internal wave breaking (as reported for other locations,

i.e. the Celtic Sea shelf, Sharples et al. 2007), but to a penetration of the nutricline into the bottom boundary layer following isopycnals, which in turn seem to adjust to the baroclinic flow of Atlantic Water along the shelf edge. The dataset presented is impressive, with a unique collection of biological, chemical and physical parameters, and the results are certainly interesting. The quality of the figures and writing are over-all good. However, the manuscript has some significant weak-points that need to be addressed before acceptance. My main comment is that, in my opinion, the results do not convincingly support the main conclusions, at least in the form in which they are presented now (see below).

General comments

1. The main conclusion that the shelf edge is an area of localized nitrate fluxes leading to new production (and increased fishing activity) is not convincingly supported by the results, at least in the way in which they are presented and discussed. If I interpret the text and figures correctly, the integrated values of chlorophyll and primary show a distinct cross-shelf distribution, being minimum close the shelf edge (Page 11, line 6, Figure 7). The authors must explain and discuss why this happens and how this relates to their statement that new production and transfer towards higher trophic levels is enhanced at the shelf edge. I could understand that larger NP may not necessary result in larger PP but this needs to be discussed at least. From figures 7 and 8 it is not entirely clear if f-ratios are larger there because primary production rates are relatively low or because nutrient fluxes are larger. Reporting mean/median values of PP and FNO3 at the different regions (shelf, shelf edge, Norwegian Trench) in Figure 9 would definitely help. Also, a statistical analysis/error assessment would be needed to show that the differences between regions are significant, particularly in the case of turbulent nitrate fluxes, which are highly uncertain due to the chaotic nature of turbulent mixing. Hence, the presentation and discussion of the results need to be significantly improved. Until then, the title of the manuscript ("Turbulence measurements suggest high rates of new production over the shelf edge in the north-eastern North Sea during

summer") is not justified.

Response: We would like to thank the reviewer for a careful positive review with constructive comments and criticism of our manuscript. We reply to all the comments below: a) We have added a new panel of the spatial distribution of the nitrate flux in figure 7c. b) We have added more text explaining the chlorophyll and PP distribution around the shelf edge region. In the end of section 4.3 we add: "Thus, the tendency to increased chlorophyll and PP on either side of the shelf edge could also be explained from the gradual build-up of biomass as nutrients are transported away from the shelf-edge region by isopycnal mixing. Alternatively, the tendency to low values could also be explained by a larger grazing pressure above the shelf edge. Thus, a full explanation of the tendency to low chlorophyll and PP above the shelf edge area cannot be determined from these data." c) To clarify that it is primarily an increase in nitrate fluxes that explains the variation in the f-ratios, we have added a new figure 7c where the nitrate flux is shown. From this figure, it can be seen that the elevated nutrient fluxes are in accordance with the distribution of the f-ratios. d) we have now assessed the distribution of the nitrate fluxes in a more quantitative manner by including Table 2, where the median values are related to the distributions of the parameters across the shelf edge. The table supports the discussion of the distributions shown in figure 7 and 8. Based on the statistical analysis in Table 2, we have now modified the depth range of the shelf edge zone to be between 80 - 130 m. The nitrate flux is largest in the depth interval representing the shelf edge (80-130). However, the average values are not significantly larger than in the other two depth intervals. This does not rule out that the fluxes are larger at the shelf edge, because of the transient nature of short-term term mixing events. We now show and analyse data from a time series station at the shelf edge at Tr4 (Fig. 8) where temporal changes associated with tides and other short-term variations can be seen. Thus, even though the average value is not significantly higher at the shelf edge the fact that the largest fluxes are observed in this area supports the interpretation of figure 7 and 8.

Reviewer 2: 2. The mechanisms that cause the nutrient fluxes to be larger at the shelf edge are not sufficiently discussed. In particular, it is not clearly shown if larger nitrate fluxes are related to enhanced turbulent dissipation, reduced stratification or enhanced nitrate gradient. I think this is important for the interpretation of the NP dynamics in the area. Additional figures showing the nitrate and buoyancy frequency distribution would help. In the discussion (Section 4.3), the authors point out that the dynamics of the shelf edge in the study area is different from similar locations, where enhanced turbulence and nutrient supply is sustained by internal tide dissipation at the shelf edge (eg. Sharples et al 2007). The authors say that they have carried out some time-series measurements to study the internal wave activity at the shelf edge and they have not found any signal of enhanced mixing (why not show this data at least as Supplementary Information?). They suggest, instead, that the enhanced nutrient fluxes at the shelf edge relate to the deepening of the nitracline at the shelf edge, reaching the bottom boundary layer. This deepening would be related to the baroclinic flow of the nutrient-rich AW at the shelf slope. This could be a very interesting point of the manuscript but it needs to be more clearly demonstrated with data allowing for a more thorough characterization of the site's dynamics, i.e. some current measurements (if available), or at least discussed in more depth with additional support from the literature.

Response: We referred only to the mixing of nutrients in our comments on the time series stations. We have clarified that we cannot identify the specific mixing process, thus we cannot disregard the influence from internal waves or tides based on the present data set. After our references to the papers of Sharples, Burchard and Rippeth in the start of section 4.3 we clarify this by adding: "Mixing associated with wind and tides (e.g. Burchard and Rippeth, 2009) as well as breaking internal waves (Sharples et al, 2007; 2009) has been shown to be important for vertical nutrient fluxes in shelf areas. The specific physical processes behind increased turbulent mixing cannot be identified from the present data set. Measurements on the time series station at the shelf edge showed that elevated mixing occurred in phase with the tidal energy input but also that additional energy sources likely contributed to the elevated mixing, e.g. energy from

non-tidal currents. Short term variability associated with advection of ambient water masses was also observed. This could possibly be related to sub-mesoscale eddies or other transport processes occurring below the pycnocline. The time series station T2 at Tr4 showed an important feature where mixing associated with the bottom boundary layer increased and intersected the bottom of the euphotic zone. Thus, the combined effect from a deep nutricline and elevated mixing provide a mechanism for increased diapycnal nutrient fluxes along the shelf edge."

We have added a new figure from the time series station at the shelf edge on Tr4 which includes the distribution of the buoyancy frequency. N2 is also now shown in figure 6. The nitrate concentration has now been included in figure 2d where the distribution along Tr4 is shown. Nitrate concentrations at stations at the shelf edge are also shown in figure 6.

Reviewer 2: 3. Lack of important information: the authors have omitted some relevant information in the methods section and others (see specific comments). Also, at least two figures, which are very relevant for the scientific content of the manuscript, must be added: (1) the distribution of nitrate concentration along at least one of the transects and (2) a comparison of the modeled vs. measured PP values at in situ conditions for the stations where they are available.

Response: We have added the information to the Methods section and Supplementary information, as described below. (1) We now show the nitrate distribution along Tr4. (2) we explain the calculation of PP in more detail and include references to two papers on uncertainties related to photosynthetic parameters. The photosynthetic parameters are derived from laboratory experiments and, therefore, the uncertainty on these parameters are the relevant measure in this context. This is discussed in detail in section 4.2. To further clarify this issue, we have now included the following paragraph in the section. "Primary production estimates at individual sites are dependent upon the value for maximum rate of photosynthesis (PBmax). However, PBmax (and all other measured photosynthetic parameters) represent the physiological condition of the phytoplankton

community at the time of sampling. This means that PBmax may vary as a function of time of sampling (Richardson et al., 2017) or during different light conditions (e.g., photo-inhibition). Normalisation of the photosynthetic parameters with chlorophyll also represents an uncertainty in the PP estimates at individual stations as, for example, division with low chlorophyll values (e.g. some surface values were ∼0.1 mg chl m-3) may result in large uncertainty of the normalised values due to relatively large absolute errors. This uncertainty error has been shown to potentially have a significant impact on the estimation of photosynthetic parameters (e.g., Kumari, 2005; McKee et al., 2015). Finally, the fact that photosynthetic parameters were determined from incubations carried out on only one water sample from each sampling depth represents a source of uncertainty with respect to the estimates of PP at individual stations. Therefore, in order to compare PP estimates from the stations we sampled, we applied average values (median for all stations) of photosynthetic parameters in the surface layer (5 m) and in the SCM in the calculation of PP. The uncertainties associated with the photosynthetic parameters are further considered in the Discussion. Surface values were assumed to represent the photosynthetic parameters in the upper 10 m and average values from the SCM were assumed to represent the parameters for the water column below 10 m."

Reviewer 2: 4. Structure: The structure of the manuscript is not always linear. I suggest some reorganization of the text/figures (eg. see specific comment 17)

Response: We have considered the specific comments in 17.

Reviewer 2:Specific comments 1. Abstract: "Estimated nitrate fluxes due to turbulent vertical mixing into the euphotic zone were up to 0.5 - 1 mmol N m-2 d-1 over the shelf-edge (f-ratios > 0.1) while values of < 0.1 mmol N m-2 d-1 were found in the deeper open area north of the shelf-edge." If this refers to figure 8, those numbers are not easy to read from this figure. A logarithmic scale must be used. Mean/median values (and uncertainties) could be reported in Figure 9.

Response: The maximum nitrate flux is now also shown in figure 7c and the large values in the shelf area are seen by the color shading (yellow-red) while the low values in the deeper open area appears as blue. We report median values in the new Table 2.

Reviewer 2: 2. Section 2.2. Important information is lacking in this section. What was the final vertical resolution of the TKE dissipation rate? How many casts were performed at each station?

Response: The resolution was ~3 m and this information is now added to the Methods section. In general, one cast was made at the relatively closely spaced stations and average values from the two shear sensors were reported. A quality criterion is, in the revised manuscript, now placed on the measurements used in the analyses. The derivation of this criterion is explained supplementary information, figure S1 and described in the Methods section.

Reviewer 2: 3. Section 2.3. How many nutrient and chlorophyll profiles/samples were analysed? "In some cases/At some stations" are very vague expressions. What was the intended horizontal and vertical resolution for nutrients? How were the sampling stations chosen?

Response: We have added the following to section 2.3: "In total, 649 water samples were analyzed for nutrients.". The distribution of nutrient samples in the vertical is described in the Methods section and it can now also be seen in figure 2d along Tr4 (shown with small bullets). The number of chlorophyll samples used for the calibration is reported in the end of the section: (n=205).

Reviewer 2: 4. Section 2.4, Page 5, lines 11-12. The goodness of the fits to eq. (3) is not sufficiently demonstrated. The authors should provide any measurement of this goodness and/or some plot of the data and fitted lines.

Response: We apply the model in Eq. 3 for calculating PP. The parameters are determined from the incubations and the three parameters at the two depth levels are shown

in Table 1. Uncertainties associated with the photosynthetic parameters are discussed in section 4.2 and we also added a new paragraph about this, cf. our response to the general comment no. 3, above. In the end of Section 4.2, we evaluate the uncertainty from the photosynthetic parameters and assess the associated uncertainty on the PP-value to be +/-30%.

Reviewer 2: 5. What is the difference between PBmax and PBmax*?

Response: In a simple PP-model without photoinhibition, then PBmax becomes the asymptotic maximum value of PP in a PP-Irradiance diagram (e.g. following the traditional simple PP-model of Webb et al (1974), Oecologia, 17, 281-91). However, the maximum of the PP-Irradiance curve is not exactly at PBmax when photoinhibition is included, cf. Eq. 3, and, therefore, PBmax* is used to quantify the maximum of the curve. We have clarified this in the parenthesis in the end of the paragraph.

Reviewer 2: 6. FNO3 calculation. If I understood correctly, the FNO3 fluxes into the photic zone at each station are reported as the maximum of the FNO3 across the nitracline. Thus, the reported fluxes are the result of a point by point multiplication of "measured" Kv values and calculated NO3 gradient. Kv has generally a patchy distribution in space and episodic in time, so that the fluxes calculated in this way my contain spurious values. How did the authors deal with this? Did they apply any averaging to the "measured" Kv values? How many casts were done at each station? The robustness of the FNO3 calculation must be assessed through a more thorough error analysis.

Response: FNO3 is calculated as the maximum flux into the euphotic zone. It was found that using the nitracline specifically as a relevant boundary may lead to an underestimate of the flux into the photic zone. We analyze this finding and document that the maximum flux is, in general, located a little deeper than the nitracline. We agree with the considerations about Kv and vertical variation and this motivated the use of Eq. 5. There are still fluctuations but these are not influenced by uncertainties associated

with Kv and division of low values of N2.

Reviewer 2: 7. Page 7, lines 7-23, Fig. 2. The authors could identify the different water masses with a text label in Figure 2. Also, the authors may outline the main circulation patterns of the different water masses in Figure 1 and provide some geographic indications (name of the countries and some topographic) features to facilitate the orientation of the reader.

Response: We have added the names of the countries to fig 1b as suggested by the reviewer. We find that water masses are well represented in figure 1a and figure 4. They could be added to figure 2a, as suggested by the reviewer, but it would make the figures less harmonic, in our opinion, so we have avoided adding more information in the figure. The water masses can be identified from the TS-relations described in the text. We also chose not to add more information about the general surface circulation in figure 1 as these figures already contain many layers of information.

Reviewer 2: 8. Page 7, lines 7-23, Page 8 lines 1-23. Though extremely relevant for the study and extensively described in these lines, nitrate distributions are not shown in the manuscript. The authors must at least include the nitrate distribution of transect 4 in Figure 2.

Response: We have followed the suggestion by the reviewer and added the nitrate distribution along Tr4 in figure 2. All nitrate measurements are also shown in figure 4a-c and observations are also now shown in figure 6 at four stations along the shelf edge.

Reviewer 2: 9. Page 8. I don't believe that adding a new subsection (3.2.1) is necessary here.

Response: We have removed the heading of subsection 3.2.1 and the section is now simply a part of section 3.2

Reviewer 2: 10. Fig. 3: there is some overlapping between the red circles and orange

squares and in some cases it is difficult to know whether some points are lacking or hidden. You could use different sizes

Response: We have increased the size of the orange squares so it is easier to see the overlapping points.

Reviewer 2: 11. Figs. 2. and 4. In the methods section, the authors say that sections 2 and 4 were repeated to study the temporal variability. Are the distributions presented in Figs. 2 and 4 a mean of the different occupations, or how were they calculated?

Response: We carried out repeated measurements at two fixed geographic locations on Tr2 and Tr4, respectively, (referred to here as "time series stations"). Both data sets are included in the revised manuscript. We apply measurements from time series 1 (107 casts in three sequences with about 3 min intervals and over 22 hours down to 62 m) to analyze the statistical significance of the epsilon-values (described in this revised mansucript in Methods and Supplementary material and shown in Figure S1). In addition, we include the time series data from the shelf edge at Tr4 over a 36h period in the new Figure 8 to show the temporal variability and the influence from tidal currents on mixing in the bottom boundary layer.

Reviewer 2: 12. Fig. 6. The vertical distribution of FNO3 is very difficult to appreciate in this figure because it follows the logarithmic variability of Kv. The authors may use a log-scale for FNO3 too (also in Fig. 8). The largest FNO3 are shown for the lower boundary layer, due to larger values of the diapycnal diffusion coefficient. The nitrate gradient however is very weak here, so I doubt whether these large fluxes would actually different from zero if the uncertainties in the nitrate gradient calculation and Kv were accounted for. Error bars should be added to the nitrate flux.

Response: Figure 6 has now been expanded so it shows four stations taken within a few hours across the shelf edge at Tr4. In this area, there is a vertical gradient in nitrate, as seen in Fig 6 and also from the new figure 2d. Thus, increased fluxes here are due to the combined effects of increased mixing and nitrate gradients. We now

describe uncertainties of epsilon in the supplementary information and uncertainties in nitrate determination are described in the Methods section. However, from the large nitrate gradient and the relative increase in the epsilon-values in the deeper part of the water column (i.e. below ∼40 m in fig. 6) it can be seen that the increased nitrate flux is significant. We have not attempted to show error-bars on the figures because they already contain several curves.

Reviewer 2: 13. Section 3.3. I would have expected to find a description of the spatial distribution of the nitrate fluxes here similar to previous sections.

Response: The new figure 2d contains the nitrate distribution along Tr4. These are described in the text at the end of section 3.1 along with a description of the other panels in figure 2. The nitrate fluxes are now shown in figure 7c.

Reviewer 2: 14. Section 3.5 / Figure 7. Vertically integrated quantities (Chlorophyll and PP) are reported in this section/figure. However, I could not find the integration depth in the manuscript. I guess that they have been integrated in the euphotic zone but this should be specified.

Response: This is now specified in the start of section 3.5: "The vertically integrated chlorophyll in the euphotic zone (50 m) ..."

Reviewer 2: 15. Section 3.5, Page 10 Line 19. How do the extrapolation with equation 3 compares with measured PP at local conditions at the locations where direct measurements are available? I suggest to add a new figure where modeled and measured values are compared.

Response: We have added more information on the photosynthetic parameters. This is described in our response to the general comment no. 3 above. We argue that the photosynthetic parameters are the relevant quantities to evaluate because these parameters are directly derived from the incubation experiments. Therefore, we have not included a new figure as suggested by the reviewer.

Reviewer 2: 16. Figure 7c and text. There is some overlapping of the color dots here and it is difficult to see whether there is a clear background tendency towards higher fratios at the shelf edge or there are only a few large values superimposed to a generally low background. How does this relate to the episodic nature of turbulent mixing? I would suggest to calculate average f-ratios for the shelf, the shelf-edge and the Norwegian Trench based on the mean (or median values) of PP and FNO3 in the different regions, instead of the point-wise calculation presented here. This numbers could be shown in Figure 9. This would also allow for a quantitative evaluation of the significance of the differences in NP between the different areas.

Response: The size of the color dots for the low values are slightly larger than for the larger values, to make it clearer to see the distribution. Regarding the episodic nature of the mixing, we expect that the spatial distribution will be influenced by this. However, because of the relatively large number of stations, we consider the distribution to reflect the general distribution of mixing in the area. We have now added a Table 2 where the parameters are calculated in different depth sections across the shelf edge, as suggested by the reviewer. The Table is analysed in section 3.6 where we have added the following:" The distributions were analysed across the shelf edge by dividing the stations into three depth ranges characterising the shallow area (50 - 80 m), the shelf edge zone (80 -130 m) and the deep area (> 130 m), respectively (Table 2). Although the shelf edge was characterised by the largest nutrient fluxes, the averaged values were not significantly higher than observed above the deeper areas. However, the depth of the maximum flux was found to be significantly deeper ($\sim$43 m) above the shelf edge than in the deeper area ($\sim$32 m). This can be explained by the significantly deeper nutricline at the shelf edge ($\sim$35 m) than observed above the deeper area ($\sim$27 m). Distributions of vertically integrated chlorophyll and PP support that minimum values are found above the shelf edge. However the low values are not significantly different from the larger values above the deeper and shallower part of the area.". The episodic nature of mixing is considered in the final paragraph in section 3.7.

Reviewer 2: 17. Sections 3.5 an 3.6 / Figures 7 and 8: The information about the spatial distribution of PP and integrated chlorophyll-a is somehow dispersed and repeated in these two figures/sections. On the other hand, in my opinion, the description of the spatial variability of the nitrate fluxes -which seems to be a central topic of the manuscript- is insufficient. I would replace the f-ratio in Fig. 7 by the actual nitrate flux and describe its variability and drivers (changes in nitrate gradient, stratification and TKE dissipation) in section 3.3, for example.

Response: We have added a new panel to figure 7 (Fig. 7c) where the flux is shown, as suggested by the reviewer. We have moved the previous figure 8 to the Supplementary information (Figure S2) because most of the information now is contained in figure 7. However, the figure shows the quantitative distribution more precisely and it also contains information of the temporal variation in, for example, PP along Tr4.

Reviewer 2: 18. Figure 8. The location of the shelf edge is not evident at all in this figure and this weakens the authors' main point (new production is enhanced at the shelf edge). I would suggest to represent the different variables as a function of the distance to the shelf edge instead of latitude. The smooth cross shelf distribution of FNO3 and the f-ratio outlined in Figure 9 and the abstract (see first comment) is not clear in this figure due to the large short-scale variability of these quantities. I would suggest to use logarithmic scale or even add a representation of FNO3 in figure 7, report mean values in Figure 9, and remove figure 8.

Response: We have followed the suggestion about adding FNO3 to figure 7 and now also report median values in Table 2. We have moved figure 8 to the Supplementary Information as described above in lines 644-648. We have kept the reference to latitude in the figure instead of distance to the shelf edge for simplicity. The location of the shelf edge along the five transects can be seen in figures 1 and 3.

Reviewer 2: 19. Figure 9. This figure is promising but it definitely needs more information. I would add mean values of primary production and nitrate fluxes (at least).

From Figures 7 and 8 it is very difficult to know if the larger f-ratios at the shelf edge are mostly due to enhanced nitrate fluxes or reduced primary production in this area. How were the f-ratios calculated, are they mean/median values or just an estimate of their order of magnitude? This is the main message of the manuscript and the authors should provide a solid quantification (and some error assessment) of the f-ratio.

Response: Median values of PP and nitrate fluxes are now reported in Table 2. Figure 7 has been improved by the addition of the distribution of FNO3max. From this, it can be seen that it is the enhanced nitrate fluxes that explain the distribution of the f-ratios. The f-ratios were calculated as the ratio of the maximum nitrate flux into the euphotic zone (converted to units of carbon by the C:N ratio) and divided by the PP. This was done for every station and is described in the end of section 2.5.

Reviewer 2: 20. Section 4.1. This section could be much improved if a comparison between modeled and measured PP values was shown.

Response: This is described in our response to the general comment no. 3 above.

Reviewer 2: 21. Page 14, lines 10. "Finally, estimates of new production imply a conversion from nitrate to carbon and a fixed ratio may not be representative for the different communities in the area." Is not there any quantification of plankton stoichiometry in the area available to assess the validity of the chosen C:N ratio?

Response: We rely on the Redfield C:N ratio and have added the original reference to the Methods section. The C:N ratio of phytoplankton has been investigated in numerous studies and, in general, a ratio of 106:16 is a good representation of the stoichiometry of the plankton. Variation is known to occur due to various causes, for example due to varying nutrient conditions and, therefore, previous values from the area would not necessarily be a better representation than simply using the "original" C:N ratio. We do not regard this to be the most critical assumption in our analyses and we have shortened the description of various error-sources in the Discussion accordingly.

Reviewer 2: 22. Page 14, line 29-30: "Mixing from tides (Sharples et al, 2007; 2009) and breaking internal waves (e.g. Burchard and Rippeth, 2008) has been shown to be important for vertical nutrient fluxes in shelf areas." This sentence is imprecise. In Sharples et al. (2007) mixing is enhanced due to internal wave breaking (in particular to the dissipation of the internal tide) and in Burchard and Rippeth (2009) enhanced turbulence is due to the the alignment of the shear vectors induced by different sources (inertial oscillations, wind and tidal bed friction). Also, the Burchard paper is from 2009, not 2008. In general this section has great potential, but needs to be improved (see General comment 2)

Response: We have corrected the introduction to the section so now it reads:"Mixing associated with wind and tides (e.g. Burchard and Rippeth, 2009) and breaking internal waves (Sharples et al, 2007; 2009) ...."

Reviewer 2: 23. Page 15, Lines 18-26. This paragraph does not match the section heading

Response: We have added a new heading to this paragraph so it now is referred to as: "Vertical nutrient fluxes in the euphotic zone"

Reviewer 2: Technical comments 1. Page 6, line 21 and Page 9 line 10. There are too much ")"

Response: These sentences have been changed to avoid double parenthesis.

Reviewer 2: 2. Page 17, line 7. Rippith ! Rippeth

Response: This has been corrected.

Please also note the supplement to this comment:
https://www.biogeosciences-discuss.net/bg-2018-385/bg-2018-385-AC2-supplement.pdf

**Supplement:**

[revised manuscript text omitted]

Figure 2

[Figure]

Figure 6

[Figure]

**Figure 7**

33

[Figure]

**Figure 8**

**Supplementary material**

**Analysis of turbulence variation and uncertainty**

Temporal variation and uncertainty associated with the shear probe measurements were assessed at a time series station, T1, located on Tr2 (57.287 °N, 7.758 °E; 62 m deep) starting 20 July 23:25 and ending 21 July 21:44 (Fig. S1). In tota,l 107 profiles were made to ~50 m depth in three sequences during the period with typically ~3 minutes between each cast. The temporal variation showed a modest change in temperature between 10-20 m (Fig. S1a) and a relatively small dissipation of turbulent kinetic energy (TKE) at mid-depth during the first 6 hours of the measurements (Fig. S1b). The instrument was equipped with two shear probes and the dissipation of TKE ($\varepsilon$) was calculated from each of the probes. The difference between these two estimates, made across the same water volume, are analysed below to assess the uncertainty of the $\varepsilon$-values.

Samples above 15 m were disregarded in the error-analysis to avoid any influence from the movement of the ship. The relative difference between the calculated dissipation of TKE ($\varepsilon$) obtained from each of the two shear probes (i.e., $\varepsilon_1$ and $\varepsilon_2$) was calculated as: $\Delta Log_{10}\varepsilon = Log_{10}(\varepsilon_1) - Log_{10}(\varepsilon_2)$. In total, there were 1145 pairs over the 22h period with a relatively small $\Delta Log_{10}\varepsilon$ average value of -0.063 and a standard and absolute deviation of 0.23 and 0.14, respectively. We applied the absolute deviation, i.e. the more conservative estimate, as being representative of the uncertainty of the $\varepsilon$-values. The relative probability distribution of $\Delta Log_{10}\varepsilon$ showed a qualitative accordance with a normal distribution characterised by the average value and absolute deviation (Fig. S1b), although the error-distribution showed a tendency to a broader variance for $\Delta Log_{10}\varepsilon$ larger than ~0.4. This was also clear from the cumulative probability distribution (Fig. S1d) where the error-distribution deviated from a normal distribution (confirmed by a Kolmogorov-Smirnov test). We considered the largest values of $\Delta Log_{10}\varepsilon$ to indicate sources of errors which could not be directly related to the instrument but potentially associated with the measurement procedure, for example influence from the rope on the free-falling instrument (all casts were made with free and undisturbed line to the free falling instrument during the whole cast). Therefore, we applied the criterion that only measurements where $\Delta Log_{10}\varepsilon$ was less than 3 times the absolute deviation (i.e. 0.42) were considered to be acceptable and these were included in the analysis. This criterion eliminated only a small number of the $\varepsilon$-values from the data set.

Temporal variation was also considered from the time series measurements at T1. Variation of $\varepsilon$ is expected to vary due to tides, wind, breaking internal waves etc. Therefore, variations at a single time series station cannot be expected to be representative for the data set as a whole. However, the short-term temporal variation was analysed from samples of $\varepsilon$ binned in 5 m intervals and analysed over a period of 30 minutes (i.e. 11 casts) resulting in average values and absolute standard deviations of 1.6±0.6, 1.4±0.6 and 2.1±1,.0 in depth intervals between 25-30m, 30-35m and 35-40m, respectively (in units of $10^{-9}$ W kg$^{-1}$). Thus, short term variation was relatively small and temporal changes between subsequent casts were considered to have a small influence on the calculated $\varepsilon$-values. Therefore, $\varepsilon$-values were, in general, derived from a single cast between the relatively closely spaced stations, where the $\varepsilon$-value obtained by averaging the calculated value from the two probes was reported. In addition to the time series station, T1, a similar time series station (T2) located at Tr4 is discussed in the text.

**Supplementary figure legends**

**Figure S1** Turbulence measurements from time series station, T1, located on Tr2 (57.287 °N, 7.758 °E) starting 20 July 23:25 and ending 21 July 21:44. (a) Temperature (°C) and (c) turbulent kinetic energy dissipation ($\varepsilon$, W kg$^{-1}$) were measured in 107 profiles (small bullets) in three sequences during the 22h period and are shown as a function of pressure and time (Day of the Year). (b) The error-distribution ($\Delta \text{Log}_{10}\varepsilon$ in units of $\text{Log}_{10}$(W kg$^{-1}$), see text) between calculated $\varepsilon$ from the two shear probes (gray bars) and the normal distribution (green) associated with the average and absolute deviation of the error-distribution. (d) The cumulative probability of the error-distribution (black) compared with the associated normal distribution (green).

**Figure S2** Distributions along the five transects of (a) vertically integrated chlorophyll a (mg chl a m$^{-2}$), (b) primary production (mg C m$^{-2}$ d$^{-1}$), (c) nutricline depth (m), (d) maximum nitrate flux into the euphotic zone (mmol N m$^{-2}$ d$^{-1}$) and (e) f-ratio in euphotic zone. Repeated stations on Tr2 and Tr4, separated in time by about a week, are shown with bullets and open circles.

[Figure]

Figure S1

29

---

## Referee Report (RR1)

**2ⁿᵈ review of "Turbulence measurements suggest high rates of new production over the shelf edge in the north-eastern North Sea during summer" by Bendtsen and Richardson.**

The authors correctly addressed my main concerns and the revised manuscript was significantly improved. The revised manuscript includes, among others, a more thorough evaluation of the uncertainties associated with the TKE dissipation and nitrate fluxes (extended Figure 6) calculations; new panels in Figures 2 and 7 displaying the distribution of nitrate in transect Tr4, and the distribution of nitrate fluxes across the shelf edge; microstructure and CTD data from a 36-hour time-series, and a Table presenting the median of relevant parameters for the different regions. Those modifications strengthen the communication of the manuscript's main conclusions. I have only a set of relatively minor points that I would like the authors to address before the manuscript is published.

GENERAL COMMENTS:
I contend that some weaknesses remain in the communication of the main conclusions. See for example the following sentences:

*"Estimated nitrate fluxes due to turbulent vertical mixing into the euphotic zone were up to 0.5 - 1 mmol N m-2 d-1 over the shelf-edge (f-ratios > 0.1) while values of < 0.1 mmol N m-2 d-1 were found in the deeper open area north of the shelf-edge." (Abstract, P.1 L. 13-14)*

*"Nitrate fluxes were generally very low (< 0.1 mmol N m-2 d-1) in the deeper area north of the shelf edge due to low vertical mixing in the upper 50 m. However, elevated nutrient fluxes of ~1 mmol N m-2 d -1 were seen in the shelf edge zone and near the Norwegian coast. This resulted in f-ratios above 0.10 in these regions compared to < 0.02 for the remainder of the study area (Fig. 7, S28d, e)." (P. 14, L. 10-14)*

*"Significant NP was found above the shelf-edge where vertical nitrate fluxes of 0.5 - 1.5 mmol N m-2 d-1 implied f-ratios above 0.10, whereas very low nutrient fluxes characterised the open area above the Norwegian Trench (f-ratios < 0.02)." (Conclusions, P. 21, L. 15-17)*

Although the nitrate diffusive flux was as high as ~1 mmol m-2 d-1 and f>0.10 in some stations over the shelf edge area, those values seem not to be representative of the "mean state", according to the median values presented in Table 2. The median flux in the shelf edge area and the corresponding f-ratio (0.11 mmol m-2 d-1, f-ratio = 0.021) are well below the values reported in these sentences. I would suggest to rephrase the sentences to acknowledge that, although values of the f-ratio/nitrate fluxes at individual stations could be as high as 0.1 and 1 mmol m-2 d-1, those values do not correspond to the regional means. You could also include the value of the median f-ratio in table 2.

SPECIFIC COMMENTS:
- This sentence "Every cast provided two shear-measurements and, in general, there was relatively small deviation between the two shear probes. Therefore, all measurements were included in the analyses and values from the two shear-probes were averaged." (P.4, L 20) seems contradictory with the following paragraph ("This led only to the removal of a few pairs of epsilon from the dataset", P.4, Line 32)

- Although I think that I understand what the authors meant with these two sentences: " Thus, PBmax was significantly lower at the SCM, in general accordance with previous studies (see review in Richardson et al., 2016) whereas **αB showed a weak decrease with depth** (with overlapping uncertainty intervals between the two depth levels). In general, **αB has been found to increase with**

**depth** (resulting in a more efficient photosynthetic response at low light levels) and inspection of the vertical distribution showed at tendency to higher values between 15 -25 m depth (i.e. 2 - 3 e-folding depth of PAR) and lower values below 30 m resulting in a lower median value from SCM level." (P. 12, L. 15-19); they might also sound a bit contradictory and probably need some clarification.

- In P 14, Line 5: "cf. Fig. 2c,d" I think the authors wanted to refer to Supplementary Figure 2 here. Also, the description of the figure remains extensive in the text. Although it contains some repeated information, I would also agree if the authors decided to keep this figure (or a reduced version of it) in the main manuscript file. My main concern with this figure was that the cross-shelf variability of the nitrate fluxes was nuanced here, but this is now clearly illustrated in Figure 7c.

- I appreciate that the authors extended the discussion about the calculation of photosynthetic parameters and its uncertainties. However, I still would like to see how the fitted curves and the data actually look like, as they are a central point of the manuscript. Could you show some figure as example to me and, maybe, include them in the manuscript or as supplementary information?

-Figure 7 (caption): Could you add the integration depth in the Figure caption as well?

- Table 2 (caption). "...depth intervals of depth,", please remove one "depth"

---

## Author Response (AR2)

Dear Editor of Biogeosciences,

We would like to thank the two reviewers for their reviews of our revised manuscript "Turbulence measurements suggest high rates of new production over the shelf edge in the north-eastern North Sea during summer". We have addressed all the minor issues raised by the reviewers in the revised manuscript.

Reviewer #1 suggested an analysis of the temporal variability of the timeseries data shown in the supplementary material. This has to some extent already been addressed in the supplementary material but we did not describe these results in detail in the main text. Therefore, this may not have been clear for the reviewer. We now include a reference to these results in the main text. In addition, there was a reference to a table in the legend to figure 6 which led to some confusion about where these data were to be found. We have now made a simpler and more straightforward description of this.

Reviewer #1 also commented on the distribution of maximum nitrate fluxes. This information can be obtained from figures 7c and 3 and we have clarified this in the text. Finally, we acknowledge reviewer #1's interesting considerations about the role of meso-scale variability along the shelf-edge indicated by the satellite images and believe this is something that should be studied further.

Reviewer #2 had a general comment on the description of the nitrate fluxes and also that it should be clarified that references to numerical values in previous studies were temporal averages (this point was also mentioned by #1). We agree on this issue, and we have clarified this when we compare our results with earlier work. Reviewer #2 also asked for examples of the incubation results and we have added an extra figure in the supplementary material showing the results from the first three stations on the cruise where we made these incubations. We also include the results from these incubations shown in the figure legend to figure S2. We have made minor improvements of figure 1 (color scale) and figure 5 (adding small arrows above 5b) and clarified the text according to the suggestions by the reviewers.

We believe the manuscript has been significantly improved by the careful reviews and constructive comments from the two anonymous reviewers and we acknowledge this in the manuscript.

Our detailed response (in normal font) to the reviewer's comments (shown in italics) is enclosed below and a version of the manuscript with "track changes" is also enclosed for your reference.

I am looking forward to hearing from you.

Kind regards,
Jørgen Bendtsen

*Reviewer #1*

*Suggestions for revision or reasons for rejection (will be published if the paper is accepted for final publication)*
*Thank you to the authors for a well-structured response to the earlier review.*
*Most of the queries I had have been answered/clarified. I have just a few final points:*

*1. I still do not fully understand how the individual, single-cast turbulence measurements (or averages based on a handful of closely-spaced single profiles) compare to the tidal variability seen at the time series station. Tides are clearly important in determining the variability in mixing, but this has not been fully explained. In Fig. S1 there appears to be sufficient data to calculate an average turbulence profile with a measure of variability (e.g. 95% confidence intervals using a bootstrap technique on the profiles). Then comparing average profiles and confidence intervals based on a handful of closely-spaced stations elsewhere in the survey would be useful in indicating if there were significant contrasts in mixing outside of the tidal variability. This is different to the analysis done on the paired shear probes (which is a nice, robust QC of the data, but doesn't tell us about tidal variability).*

Response:
We include an analysis of the temporal variability observed at time series station T1 described in the supplementary material. The variability of the time series is analysed in vertical bins of 5 m intervals in the first part of the time series, i.e. the first 11 casts over 30 minutes, to provide information about the temporal variability between subsequent casts. It is found that the variability (ranging between 1.4 - 2.1 10-9 W kg-1) is well below the range observed in the data set. This result supports our argument that the individual casts along the transects are representative for the conditions when considered over a similar time period. From the current data set, we cannot analyse the general influence of tides along the transects and we do not make any conclusions about the influence from tidal currents, although we discuss their likely influence on mixing in the area (for example in relation to Figure 8). The suggestion by the reviewer is partly covered by the calculation in the supplementary material and we have added a more specific reference to this in the parenthesis in the following sentence (p4, l27): "Short term variation at T1 was also found to be relatively small and temporal changes between subsequent casts were considered to have a little influence on the calculated ε-values (time series analysis in Supplementary material)".
However, we cannot address the influence from tidal currents on the spatial distribution of nitrate fluxes from the present data set. We show that tidal currents likely play an important role for the temporal variability at time series station 2 (Fig. 8) but we also discuss that other sources, e.g. meso-scale variability, may be important for mixing along the shelf edge. Considering the transient nature of mixing processes, we find it quite interesting that increased nitrate fluxes are consistently located along the shelf edge. This is not a proof that the shelf edge is the only source of new production in the area but, considering the length of the cruise and number of stations, we find that this pattern is a strong indication that the area plays a very important role for biological cycling in the area.

*In the revised manuscript (section 3.7, page 13) it is stated that turbulence measurements were not made at night. Fig. S1 suggests they were made at night – assuming the DoY 202.0 is midnight. In Fig. 8 it looks like data were indeed not collected for a short time at night.*

Response:
We have added "... no turbulence measurements were made at night at this station ...", to specify and clarify that there were no measurements during night "at this station" (p13, l25). The first timeseries station, analyzed in the Supplementary material, included measurements during the night.

*On page 14, line 10 – I do not understand why the observation that turbulent dissipation was similar to tidal energy dissipation (1st sentence) then suggests that there must be another source of turbulence (2nd sentence). To me the 2nd sentence simply does not follow on from the claim in the 1st sentence.*

Response:
We have clarified the logic in this sentence. We have removed "This suggests that ..." in the following sentence that now reads: "Additional energy for turbulence may also be provided from non-tidal currents along the shelf edge ...".

*Looking at Fig. 5 it does look like additional turbulence in the bottom boundary layer over the shelf edge. Is the claim that this turbulence is not tidal? If not, are there any published reports of slope currents or eddies that could be cited as suggestive of the source of the mixing?*

Response:
We cannot (and do not) claim that the source is not tidal currents. From the data set, we cannot say what the source of energy is for the mixing in the boundary layer. However, we suggest likely energy sources could be tides or currents along the shelf edge. In addition, we describe previous studies in the area in the Introduction where increased meso-scale variability was identified in the region (e.g. the reference to Røed and Fossum, 2004). We are not aware of published reports of slope currents besides the more general descriptions of the circulation pattern in the area obtained from model studies (e.g. described in the references to the four model studies in the Introduction).

*In the supplementary material the vertical turbulence profiles are described as being "3 minutes" apart. But this is not the temporal resolution of the profiles. What is the typical time interval between the start of each profile (i.e. the period of the turbulence profiler yo-yos)?*

Response:
At this time series station, the interval was in fact about 3 minutes. The profiler was lowered to about 50 m and the following cast was repeated when the profiler was brought to the surface again. This operation took about 3 minutes and, in total, 107 casts at the station were made (three persons were continuously in action during this operation).

*2. On page 17 it is now more clearly articulated than the relatively strong turbulent flux of nitrate is a consequence of (1) deeper nutricline and (2) "elevated" mixing. But it is not clear that the increased nutrient flux is at the shelf edge. It might be shown in Fig. 6, but I cannot work out where the profiles in Fig. 6 came from. The caption refers to "summarised in the tables (e-h)". I assume e-h refers to the profiles in Fig. 6, but which "tables"? Neither of the tables in the manuscript have location information. The nitrate flux has a clear peak in profile g, associated with a peak in turbulent dissipation, but it is not clear where on a transect this was. Can the positions of the profiles be marked on one of the earlier sections? Alternatively does this "elevated" shelf edge mixing rest on the evidence of the sections in Fig. 5 (though the relationship to the nutricline is not obvious)?*

Response:
This has been clarified. The "tables" referred to are written as inlets in the figures e-h. Along with profiles from each station, the lower figure also contains information about latitude (to locate the station on the map), nutricline depth and maximum nitrate fluxes. We have replaced the reference to "summarized in the tables (e-h)" with "shown in the figures (e-h)" in the figure legend. This hopefully, clarifies this issue. In addition, we now show the locations of the four stations by arrows

in fig. 5b. The latitudes of the stations shown in the figures can also be related to the maps (figs. 1, 7). The figure legends for figs. 5 and 6 have been modified accordingly.

*In general avoid the word "elevated". It is ambiguous – do you mean "increased" or "raised upward in the water column"? I'm assuming "increased"….better to make this clearer throughout.*

Response:
We have clarified this and modified the text accordingly.

*3. Line 15, page 17. The largest nitrate fluxes of 1 – 1.5 mmol m-2 day-1 are stated (with reference to Fig. 7), but I cannot see the parameters behind these in any of the data presented. Table 2 shows mean fluxes (it does not clearly say where from in the caption), which are very low, and the maximum value I can see in Fig. 6 (g) is about 0.3 mmol m-2 day-1. There only are a handful of points in Fig 7 with these high values of nitrate flux, so without any other information (e.g. profiles of nitrate and turbulence) there is nothing to support the underlying reasons for these fluxes - they could easily be interpreted as outliers. In other words, what is the link between these high flux points, the patterns of turbulence in Fig. 5 and the flux profiles in Fig. 6? Are you attributing these high fluxes to the increased boundary mixing and deeper nutricline idea (see also point 5 below….)?*

Response:
We have modified the reference to the values above the shelf-edge so the reference to figure 7 now reads (p13, l11):"However, the largest nutrient fluxes were seen in the shelf edge zone and near the Norwegian coast. This resulted in f-ratios above 0.10 in these regions compared to < 0.02 for the remainder of the study area (Fig. 7, S2)." We have also clarified that the values in Table 2 are based on the whole data set by adding "…of all data ..." in the caption. The information about the fluxes can be found in figure 7c where the spatial distribution of the maximum flux into the euphotic zone is shown. The vertical distribution of the maximum fluxes along all transects can be seen in figure 3 (orange squares) and the vertical distribution of vertical mixing rates is presented in figure 5 along transect 2 and 4. Finally, the nutrient distribution is shown along transect 4 in figure 2. So, all the pieces of information necessary for understanding the spatial distribution of the nitrate flux are contained in these figures. Figure 6 shows an example of four stations measured within a few hours of each other where all the components in the calculations are shown. We do not consider the measurements along the shelf-edge as outliers and We would also like to emphasise that we do not consider the specific values as being the most aspect of our results. We believe the general pattern where increased nitrate fluxes  are located along the shelf-edge indicates that this area has a special significance for NP. Therefore, we have modified the references to specific numerical values of the nitrate flux. We have added more information to the paragraph describing Fig. 7c to relate the results to the other figures and table 2 (Section 3.5, p12, l12): " The vertical distribution of the maximum nitrate flux was, in general, characterized by being located below the SCM (cf. Fig. 3, Table 2). However, the horisontal distribution showed a characteristic pattern where increased fluxes and f-ratios were found along the shelf-edge on Tr2-Tr5" .

*The cited values of nitrate fluxes to the SCM from other shelf studies are all daily-means, so make it clear in the text if you are comparing maximum values with daily means, or daily means with daily means.*

Response:
We have modified the text and specify that the cited values from the two studies are daily averaged values.

*4. On page 18 it is stated that the maximum nitrate flux was on average found below the nutricline depth (by 6.4 m). Given the vertical resolution of the nitrate samples, this is not significant. Also, the maximum flux is stated as being below the SCM – but if the SCM depth is defined as the maximum chl concentration, this is not surprising. The peak of typical shelf sea SCM is often coincident with the point where nitrate hits zero (within the constraints of the resolution in this study, that looks like a reasonable claim here as well), so the relevant region over which to look at the nitrate flux is generally in the low part of the SCM rather than at the peak. Nitrate reaching zero at the SCM peak then lends support to the statement that recycling must be important further up in the SCM (or, indeed, motility given the very low turbulence in the SCM).*

Response:
We describe the calculation of the nutricline depth in the first paragraph in section 2.5: "To minimise the uncertainty of the nutricline depth estimate associated with the linear interpolation between two neighbouring water samples, the nutricline depth was found by linear interpolation between the corresponding potential density anomalies ($\sigma_\theta$) of the sample depths, and the nutricline depth was then identified from the corresponding $\sigma_\theta$ in the CTD-profile. This approach is based on the assumption that the nitrate concentrations between two water samples are more closely related to water mass characteristics than linearly to depth changes, i.e. a sharp pycnocline, not resolved by the water samples, is taken into account when the nutricline depth is estimated by this method." Thus, the nutricline depth is calculated at the same resolution as the CTD-profile (the CTD sampled at 24 Hz and was bin-averaged every 0.5 m) and, therefore, the result that the nutricline depth is located, on average, 6.4 m below the SCM is significant. This is actually a quite interesting result as it implies a transport between the SCM and the maximum N-flux. We agree in the reviewer's considerations related to this about regenerated production in the lower part of the euphotic zone and the possible influence from motility.

*5. The addition of the satellite imagery is really interesting. It looks to me (comparing Figs 1 and 7) that the high values of nitrate flux are coincident with fronts in SST associated with what might be an outflow from the Baltic. I have no idea if this is coincidence or causal, but could this be tentative evidence of another source of flow and shear in the system? In relation to my point 3 above, could the mechanism underpinning these highest fluxes that you see be different from the boundary-driven turbulence plus deeper nutricline idea? The more I look at the data, the more I think there may be two mechanisms behind the pattern of nitrate fluxes: (1) shelf edge boundary mixing combined with deeper nutricline for the "typical" fluxes of 0.1- 0.3 mmol m-2 day-1, and frontal or mesoscale shear associated with the warm surface water from the Baltic driving the larger fluxes (which are patchy in the survey data, but could be quite significant given the scale of the SST signal). So the highest f-ratios might be frontal signatures. Obviously (!) don't try to open up a new area of analysis – but there may be published work on Baltic outflows to the N Sea that might help justifying this as a reasonable hypothesis in the discussion.*

Response:
This is a very interesting comment and we agree in that meso-scale variability associated with outflowing water from Skagerrak may contribute with frontal mixing in addition to mixing associated with dynamics at the shelf-edge. We have not considered this further in this study but the role of various mixing processes on the production and ecosystem structure in the area is the subject of on-going studies. Regarding previous studies, we are not aware of studies focusing on this issue in detail. There are several model studies describing the general circulation in the area (cf. our response to comment no. 1 above and the cited references) but they do not describe the outflow dynamics and mesoscale variability in detail.

**Reviewer #2**

*Suggestions for revision or reasons for rejection (will be published if the paper is accepted for final publication)*

*The authors correctly addressed my main concerns and the revised manuscript was significantly improved. The revised manuscript includes, among others, a more thorough evaluation of the uncertainties associated with the TKE dissipation and nitrate fluxes (extended Figure 6) calculations; new panels in Figures 2 and 7 displaying the distribution of nitrate in transect Tr4, and the distribution of nitrate fluxes across the shelf edge; microstructure and CTD data from a 36-hour time-series, and a Table presenting the median of relevant parameters for the different regions. Those modifications strengthen the communication of the manuscript's main conclusions. I have only a set of relatively minor points that I would like the authors to address before the manuscript is published.*

*GENERAL COMMENTS:*
*I contend that some weaknesses remain in the communication of the main conclusions. See for example the following sentences:*

*"Estimated nitrate fluxes due to turbulent vertical mixing into the euphotic zone were up to 0.5 - 1 mmol N m-2 d-1 over the shelf-edge (f-ratios > 0.1) while values of < 0.1 mmol N m-2 d-1 were found in the deeper open area north of the shelf-edge." (Abstract, P.1 L. 13-14)*

Response:
We have specified that the results show that large values were found above the shelf edge and modified the sentence accordingly to the following: "
Relatively large (up to >0.5 mmol N m$^{-2}$ d$^{-1}$ ) nitrate fluxes due to turbulent vertical mixing into the euphotic zone were found at some stations over the shelf-edge, while low values (< 0.1 mmol N m$^{-2}$ d$^{-1}$) were found in the deeper open area north of the shelf-edge. "

*"Nitrate fluxes were generally very low (< 0.1 mmol N m-2 d-1) in the deeper area north of the shelf edge due to low vertical mixing in the upper 50 m. However, elevated nutrient fluxes of ~1 mmol N m-2 d -1 were seen in the shelf edge zone and near the Norwegian coast. This resulted in f-ratios above 0.10 in these regions compared to < 0.02 for the remainder of the study area (Fig. 7, S28d, e)." (P. 14, L. 10-14)*

Response:
We have reformulated the sentence without specifying the numerical value of 1 mmol N m-2 d-1. The sentence has been changed to: ..." However, the largest nutrient fluxes were seen in the shelf edge zone and near the Norwegian coast. ...."

*"Significant NP was found above the shelf-edge where vertical nitrate fluxes of 0.5 - 1.5 mmol N m-2 d-1 implied f-ratios above 0.10, whereas very low nutrient fluxes characterised the open area above the Norwegian Trench (f-ratios < 0.02)." (Conclusions, P. 21, L. 15-17)*

Response:
We have reformulated the sentence accordingly to the following: "Significant NP was found above the shelf-edge where, at some stations, relatively large nitrate fluxes, i.e. > 0.5 mmol N m$^{-2}$ d$^{-1}$ implied f-ratios above 0.10. "

*Although the nitrate diffusive flux was as high as ~1 mmol m-2 d-1 and f>0.10 in some stations*

*over the shelf edge area, those values seem not to be representative of the "mean state", according to the median values presented in Table 2. The median flux in the shelf edge area and the corresponding f-ratio (0.11 mmol m-2 d-1, f-ratio = 0.021) are well below the values reported in these sentences. I would suggest to rephrase the sentences to acknowledge that, although values of the f-ratio/nitrate fluxes at individual stations could be as high as 0.1 and 1 mmol m-2 d-1, those values do not correspond to the regional means.*

Response:
We have reformulated the sentences above to reflect that the distributions are characterised by the large values along the shelf edge.

*SPECIFIC COMMENTS:*
*- This sentence "Every cast provided two shear-measurements and, in general, there was relatively small deviation between the two shear probes. Therefore, all measurements were included in the analyses and values from the two shear-probes were averaged." (P.4, L 20) seems contradictory with the following paragraph ("This led only to the removal of a few pairs of epsilon from the dataset", P.4, Line 32)*

Response:
This has been corrected and clarified. The sentence is a reminiscence from the first version of the manuscript and was, unfortunately, not deleted. The new quality check applied in the revised version is described in detail in the paragraph below.

*- Although I think that I understand what the authors meant with these two sentences: " Thus, PBmax was significantly lower at the SCM, in general accordance with previous studies (see review in Richardson et al., 2016) whereas αB showed a weak decrease with depth (with overlapping uncertainty intervals between the two depth levels). In general, αB has been found to increase with depth (resulting in a more efficient photosynthetic response at low light levels) and inspection of the vertical distribution showed at tendency to higher values between 15 -25 m depth (i.e. 2 - 3 e-folding depth of PAR) and lower values below 30 m resulting in a lower median value from SCM level." (P. 12, L. 15-19); they might also sound a bit contradictory and probably need some clarification.*

Response:
We have clarified this issue further by adding "significantly" to the sentence: "In general, αB has been found to increase significantly with depth ....". The sentence is a general consideration about the distribution of αB in relation to global measurements from various areas. An interesting aspect, however, is that a similar vertical distribution was reported in the cited study of Hickman et al. in the Celtic Sea.

*- In P 14, Line 5: "cf. Fig. 2c,d" I think the authors wanted to refer to Supplementary Figure 2 here. Also, the description of the figure remains extensive in the text. Although it contains some repeated information, I would also agree if the authors decided to keep this figure (or a reduced version of it) in the main manuscript file. My main concern with this figure was that the cross-shelf variability of the nitrate fluxes was nuanced here, but this is now clearly illustrated in Figure 7c.*

Response:
We prefer to keep the figure in the supplementary material because the distributions are now shown in the panels in figure 7 and figure 3 (nutricline depth). However, the figure also shows the temporal variability and, although this is not a main finding in relation to this analysis, it is a relevant piece of information.

*- I appreciate that the authors extended the discussion about the calculation of photosynthetic parameters and its uncertainties. However, I still would like to see how the fitted curves and the data actually look like, as they are a central point of the manuscript. Could you show some figure as example to me and, maybe, include them in the manuscript or as supplementary information?*

Response:
We have included examples from the first three stations in supplementary figure S2. The figure shows incubations from the surface sample (5 m) and the SCM at the three stations. The non-linear fits are shown (lines) and the associated values of the photosynthetic parameters are listed in the figure legend. Similarly good fits were obtained from the other incubation stations. We have added a reference to the figure in the Methods section and in the first paragraph in section 4.2.

*-Figure 7 (caption): Could you add the integration depth in the Figure caption as well?*

Response
This has been added.

*- Table 2 (caption). "...depth intervals of depth,", please remove one "depth"*

Response:
This has been corrected.

[revised manuscript text omitted]

**Supplementary material**

**Analysis of turbulence variation and uncertainty**

Temporal variation and uncertainty associated with the shear probe measurements were assessed at a time series station, T1, located on Tr2 (57.287 °N, 7.758 °E; 62 m deep) starting 20 July 23:25 and ending 21 July 21:44 (Fig. S1). In total, 107 profiles were made to ~50 m depth in three sequences during the period with typically ~3 minutes between each cast. The temporal variation showed a modest change in temperature between 10-20 m (Fig. S1a) and a relatively small dissipation of turbulent kinetic energy (TKE) at mid-depth during the first 6 hours of the measurements (Fig. S1b). The instrument was equipped with two shear probes and the dissipation of TKE ($\varepsilon$) was calculated from each of the probes. The difference between these two estimates, made across the same water volume, are analysed below to assess the uncertainty of the $\varepsilon$-values.

Samples above 15 m were disregarded in the error-analysis to avoid any influence from the movement of the ship. The relative difference between the calculated dissipation of TKE ($\varepsilon$) obtained from each of the two shear probes (i.e., $\varepsilon_1$ and $\varepsilon_2$) was calculated as: $\Delta Log_{10}\varepsilon = Log_{10}(\varepsilon_1) - Log_{10}(\varepsilon_2)$. In total, there were 1145 pairs over the 22h period with a relatively small $\Delta Log_{10}\varepsilon$ average value of -0.063 and a standard and absolute deviation of 0.23 and 0.14, respectively. We applied the absolute deviation, i.e. the more conservative estimate, as being representative of the uncertainty of the $\varepsilon$-values. The relative probability distribution of $\Delta Log_{10}\varepsilon$ showed a qualitative accordance with a normal distribution characterised by the average value and absolute deviation (Fig. S1b), although the error-distribution showed a tendency to a broader variance for $\Delta Log_{10}\varepsilon$ larger than ~0.4. This was also clear from the cumulative probability distribution (Fig. S1d) where the error-distribution deviated from a normal distribution (confirmed by a Kolmogorov-Smirnov test). We considered the largest values of $\Delta Log_{10}\varepsilon$ to indicate sources of errors which could not be directly related to the instrument but potentially associated with the measurement procedure, for example influence from the rope on the free-falling instrument (all casts were made with free and undisturbed line to the free falling instrument during the whole cast). Therefore, we applied the criterion that only measurements where $\Delta Log_{10}\varepsilon$ was less than 3 times the absolute deviation (i.e. 0.42) were considered to be acceptable and these were included in the analysis. This criterion eliminated only a small number of the $\varepsilon$-values from the data set.

Temporal variation was also considered from the time series measurements at T1. Variation of $\varepsilon$ is expected to vary due to tides, wind, breaking internal waves etc. Therefore, variations at a single time series station cannot be expected to be representative for the data set as a whole. However, the short-term temporal variation was analysed from samples of $\varepsilon$ binned in 5 m intervals and analysed over a period of 30 minutes (i.e. 11 casts) resulting in average values and absolute standard deviations of 1.6±0.6, 1.4±0.6 and 2.1±1,.0 in depth intervals between 25-30m, 30-35m and 35-40m, respectively (in units of $10^{-9}$ W kg$^{-1}$). Thus, short term variation was relatively small and temporal changes between subsequent casts were considered to have a small influence on the calculated $\varepsilon$-values. Therefore, $\varepsilon$-values were, in general, derived from a single cast between the relatively closely spaced stations, where the $\varepsilon$-value obtained by averaging the calculated value from the two probes was reported. In addition to the time series station, T1, a similar time series station (T2) located at Tr4 is discussed in the text.

**Supplementary figure legends**

**Figure S1** Turbulence measurements from time series station, T1, located on Tr2 (57.287 °N, 7.758 °E) starting 20 July 23:25 and ending 21 July 21:44. (a) Temperature (°C) and (c) turbulent kinetic energy dissipation ($\varepsilon$, W kg$^{-1}$) were measured in 107 profiles (small bullets) in three sequences during the 22h period and are shown as a function of pressure and time (Day of the Year). (b) The error-distribution ($\Delta Log_{10}\varepsilon$ in units of $Log_{10}$(W kg$^{-1}$), see text) between calculated $\varepsilon$ from the two shear probes (gray bars) and the normal distribution (green) associated with the average and absolute deviation of the error-distribution. (d) The cumulative probability of the error-distribution (black) compared with the associated normal distribution (green).

**Figure S2** Example of incubation results. Incorporation of carbon (photosynthesis) is shown versus PAR for the surface (5 m, bullets) and the SCM (diamonds) and non-linear best fit solutions (lines) from stations located at a) Tr1, 57.832 °N, b) Tr2, 57.480 °N and c) Tr2, 56.261 °N (cf. Fig. 1a). The following results are obtained from the (surface, SCM) at a): $P_{max}^* = (2.5, 3.2)$ [µg C h$^{-1}$], $\alpha = (0.027, 0.074)$ [µg C·(h µE m$^{-2}$ s$^{-1}$)$^{-1}$], $\beta = (0.0013, 0.0088)$ [µg C (µE m$^{-2}$ s$^{-1}$ h)$^{-1}$] and $E_{max}^* = (338,141)$ [µE m$^{-2}$ s$^{-1}$]; b): $P_{max}^* = (4.2, 2.3)$, $\alpha = (0.050, 0.057)$, $\beta = (0.0019, 0.0056)$ and $E_{max}^* = (282,130)$ (units as in a); and c): $P_{max}^* = (1.0, 1.2)$, $\alpha = (0.012, 0.019)$, $\beta = (0.0013, 0.0012)$ and $E_{max}^* = (284,219)$ (units as in a). Chlorophyll normalized values (i.e. $P^B_{max}^*$, $\alpha^B$ and $\beta^B$) are obtained by dividing with the chlorophyll a concentration at the three stations: a) (0.52, 0.95), b) (0.38, 0.82) and c) (0.26, 0.38) mg chl a L$^{-1}$.

**Figure S3** Distributions along the five transects of (a) vertically integrated chlorophyll a (mg chl a m$^{-2}$), (b) primary production (mg C m$^{-2}$ d$^{-1}$), (c) nutricline depth (m), (d) maximum nitrate flux into the euphotic zone (mmol N m$^{-2}$ d$^{-1}$) and (e) f-ratio in euphotic zone. Repeated stations on Tr2 and Tr4, separated in time by about a week, are shown with bullets and open circles.